# Enhanced activity of Alzheimer disease-associated variant of protein kinase Cα drives cognitive decline in a mouse model

Gema Lordén[1], Jacob M. Wozniak[1,2], Kim Doré[3], Lara E. Dozier[4], Chelsea Cates-Gatto[5], Gentry N. Patrick[4], David J. Gonzalez[1,2], Amanda J. Roberts[5], Rudolph E. Tanzi ®[6] & Alexandra C. Newton ®[1] ✉

Exquisitely tuned activity of protein kinase C (PKC) isozymes is essential to maintaining cellular homeostasis. Whereas loss-of-function mutations are generally associated with cancer, gain-of-function variants in one isozyme, PKCα, are associated with Alzheimer's disease (AD). Here we show that the enhanced activity of one variant, PKCα M489V, is sufficient to rewire the brain phosphoproteome, drive synaptic degeneration, and impair cognition in a mouse model. This variant causes a modest 30% increase in catalytic activity without altering on/off activation dynamics or stability, underscoring that enhanced catalytic activity is sufficient to drive the biochemical, cellular, and ultimately cognitive effects observed. Analysis of hippocampal neurons from PKCα M489V mice reveals enhanced amyloid-β-induced synaptic depression and reduced spine density compared to wild-type mice. Behavioral studies reveal that this mutation alone is sufficient to impair cognition, and, when coupled to a mouse model of AD, further accelerates cognitive decline. The druggability of protein kinases positions PKCα as a promising therapeutic target in AD.

Alzheimer's disease (AD), the most common neurodegenerative disorder in elderly individuals, is characterized by degeneration of synapses, neuronal death, and ultimately, a reduction in the size of brain regions involved in learning and memory[1]. In addition, AD brains are distinguished by the presence of neurofibrillary tangles and extracellular amyloid-β (Aβ) plaques, which lead to cognitive impairment[2–4]. However, the molecular mechanisms underlying AD remain elusive. Currently, one of the strongest hypotheses linked to AD development is the abnormal accumulation of the Aβ 42 peptides, produced by the improper processing of the amyloid precursor protein (APP) by β- and γ-secretases[5,6]. Specifically, soluble Aβ oligomers

are considered to be the neurotoxic species that initiate the disease and its accompanying symptoms. The Aβ plaques, however, are comparatively inert, but may function as reservoirs of the diffusible oligomers[7]. APP and presenilin genes (PSEN1 and PSEN2) were among the first genes shown to have variants associated with AD and were crucial to developing the amyloid cascade hypothesis and establishing the association between misprocessing and deposition of Aβ plaques and AD development[8–11]. Although these variants account for a relatively low percentage of the cases of AD, their identification paved the way to explore the complex genetics associated with the disease[12]. Polymorphisms in APOE were also established early on as being a

[1]Department of Pharmacology, University of California San Diego, La Jolla, CA 92093, USA. [2]Skaggs School of Pharmacy and Pharmaceutical Sciences, University of California San Diego, La Jolla, CA 92093, USA. [3]Center for Neural Circuits and Behavior, Department of Neurosciences, University of California San Diego, La Jolla, CA 92093, USA. [4]Section of Neurobiology. Division of Biological sciences, University of California San Diego, La Jolla, CA 92093, USA. [5]Animal Models Core Facility, The Scripps Research Institute, La Jolla, CA 92037, USA. [6]Genetics and Aging Research Unit, McCance Center for Brain Health, Department of Neurology, Department of Neurology, Massachusetts General Hospital and Harvard Medical School, Charlestown, MA 02129, USA. ✉e-mail: anewton@health.ucsd.edu

strong risk factor for AD[13–15]. Whole-genome sequencing efforts are identifying other genetic variants associated with AD risk, including polymorphisms in genes such as *TREM2*, *PLCG2*, and *ABI3* among others[16–20]. In a recent search for rare functional variants associated with AD, analysis of whole-genome sequencing data from 410 families of affected and unaffected siblings from the NIMH cohort identified variants in PKCα. One of these variants, M489V (rs34406842, minor allele frequency of 0.00095 in gnomad), was present only in affected members and no unaffected members of 4 families, cosegregating with AD affection status[21]. The high druggability of protein kinases poises PKCα as a potential target in AD therapies.

PKCα belongs to the Ca²⁺- and diacylglycerol (DG)-dependent class of PKC isozymes referred to as conventional PKC. These Ser/Thr kinases transduce signals from receptor-mediated hydrolysis of membrane phospholipids, which generates their activators, $Ca^{2+}$ and DG[22–24]. Conventional PKC isozymes play critical roles in maintaining cellular homeostasis, where their finely tuned activity regulates the balance between cell death and survival. They are primed by a series of ordered phosphorylations required for them to adopt a stable, auto-inhibited conformation poised to respond to second messengers rapidly and reversibly. Aberrant PKC that is not properly autoinhibited is shunted to degradation by quality control pathways[25]. Although PKC has historically been assumed to be oncogenic, recent analysis has reframed PKC isozymes as having tumor-suppressive roles. Notably, cancer-associated mutations are generally loss-of-function, and elevated protein levels of PKC isozymes confer improved survival for many cancers[26,27]. For this reason, inhibition of PKC in cancer has been unsuccessful, and, in some cases, worsened patient outcome[28]. The identification of activity-enhancing variants of PKCα that co-segregate with AD[21] opens the possibility that this disease could benefit from repurposing PKC inhibitors originally used in cancer clinical trials.

In marked contrast to cancer, gain-of-function mutations in PKCα have been shown to co-segregate with AD[21]. The identification of highly penetrant germline variants in PKCα in families with LOAD highlighted deregulated PKC function as potentially causative in AD[21]. Consistent with the involvement of PKC in the pathology of AD, unbiased phosphoproteomics studies have identified augmented phosphorylation of PKC substrates, including myristoylated alanine-rich C-kinase substrate (MARCKS), as one of the main events in AD development[29,30]. Additionally, electrophysiological studies have established that PKCα is necessary for Aβ-dependent synaptic depression, by a mechanism that requires the PDZ ligand of this PKC isozyme[21,31]. This PDZ ligand targets PKCα to the scaffolds PSD95, SAP97, and PICK1, with this latter scaffold also being necessary for Aβ-dependent synaptic depression[31–33]. All AD-associated variants in PKCα described to date are gain-of-function[21,34]. Biochemical analysis of one variant present in four unrelated families, M489V PKCα, reveals that the Met to Val substitution in the activation loop increases the intrinsic catalytic rate of the enzyme by ~30% without affecting stabilizing autoinhibitory constraints, and consequently, evades the cell's homeostatic degradation of aberrantly active PKCα[34]. Taken together, these results support a model in which the activity of PKCα at post-synaptic scaffolds mediates the effects of Aβ, with enhanced activity contributing to the pathology of AD. Determining whether AD-associated mutations in PKCα are sufficient to drive the pathology of AD would inform on whether PKCα inhibition is a potential therapeutic strategy in AD.

In the present study, we used genome editing to introduce the M489V gain-of-function variant into endogenous PKCα in order to determine whether this single amino acid change, modestly enhancing kinase activity, is sufficient to drive the pathology of AD in a mouse model. Biochemical, phosphoproteomic, electrophysiological, and behavioral studies revealed that elevated PKC activity conferred by the presence of this AD variant leads to increased phosphorylation of PKC substrates in the brain, neurite degeneration, enhanced Aβ driven synaptic depression, and cognitive decline, which is more evident and

more rapid on the background of a transgenic mouse model of AD. These data establish that enhanced PKCα activity is sufficient to drive cognitive decline in a mouse model and support inhibition of PKCα as a potential therapeutic approach in AD.

# Results

## Altered brain phosphoproteome in M489V PKCα mice

PKCα M489V is an AD-associated mutation that has been shown to be more catalytically active than wild-type (WT) PKCα by a mechanism that does not compromise its stability, allowing it to evade the cell's homeostatic down-regulation of aberrantly active PKCα[34]. To identify phosphorylation events that occur in the brain as a result of increased PKCα activity, brains from C57BL/6 mice harboring WT and the M489V mutation in PKCα were isolated at 3 months of age and subjected to phosphoproteomic analysis. Briefly, proteins were extracted and digested from brain lysates and phosphopeptides were enriched with $TiO_2$. Unenriched peptides and phosphopeptides were labeled with TMT 10-plex reagents, and analyzed by liquid chromatography-MS2/MS3 (LC-MS3) to quantify the proteome and phosphoproteome (Fig. 1a). We quantified 10,208 phosphopeptides per sample representing 1899 proteins with a false discovery rate (FDR) of <1% at peptide and protein level (Dataset S1). The median within sample coefficient of variation was ~20% on average, indicating minimal variance (Supp Fig. 1b). The phosphosite distribution was 80.98% phosphoserine (pS), 17.00% phosphothreonine (pT), and 2.02% phosphotyrosine (pY), with most of the peptides phosphorylated on single (68.32%) and double (26.33%) sites and fewer on triple (4.64%) and quadruple (0.71%) sites (Supp Fig. 1c, d).

The presence of the M489V PKCα variant induced significant changes in a variety of phosphoproteins (Fig. 1b and Supp Fig. 1f). Specifically, M489V PKCα induced significant changes in a total of 829 phosphopeptides. Of those, 430 peptides from 270 unique proteins had increased phosphorylation (red) and 399 peptides from 261 unique proteins had decreased phosphorylation (blue) (Fig. 1b; Supp Table 1 presents the top 25 peptides whose phosphorylation increased in PKCα M489V brain). Among the peptides with increased phosphorylation in the brains of mice harboring the M489V variant of PKCα, we found increased phosphorylation of Ser156 and Ser163 from MARCKS, a *bonafide* PKC substrate (Fig. 1b, c). Increased phosphorylation of MARCKS, in particular, is consistent with the previously reported increased catalytic activity of purified PKCα M489V compared to WT[34], an increase also observed for PKCα immunoprecipitated from brain lysates of the M489V mice (Supp Fig. 1g). Importantly, immunoblot analysis of brain lysates from littermates of the mice used for mass spectrometry analysis has previously established that the amount of PKCα protein did not change in brains from WT and M489V mice[34] (see also Supp Fig. 1h), consistent with our biochemical studies which demonstrated that the M489V mutation in PKCα does not alter the steady-state levels of the protein[34].

Next, we evaluated the changes in the phosphoproteome caused by the presence of PKCα M489V in heterozygosity or in homozygosity using k-means clustering of the phosphopeptides whose phosphorylation changed significantly among the three groups (Supp Fig. 1e). The PKCα M489V-induced phosphoproteome separated into four different clusters, with cluster 2 (C2) and cluster 3 (C3) displaying the peptides whose phosphorylation decreased or increased, respectively, in a gene-dosage dependent manner (Fig. 1d, e, Supp Fig. 1e). To explore the biological functions associated with the enhanced kinase activity derived from the AD variant in PKCα, we used gene ontology (GO) enrichment analysis to compare the cellular compartments represented in C2 (phosphorylation of substrates decreased in a gene-dosage dependent manner) and C3 (phosphorylation increased in a gene-dosage dependent manner) (Fig. 1d, e). Post-synaptic density, synapse, cell junction, and post-synaptic membrane were significantly enriched in the group of peptides whose phosphorylation decreased in

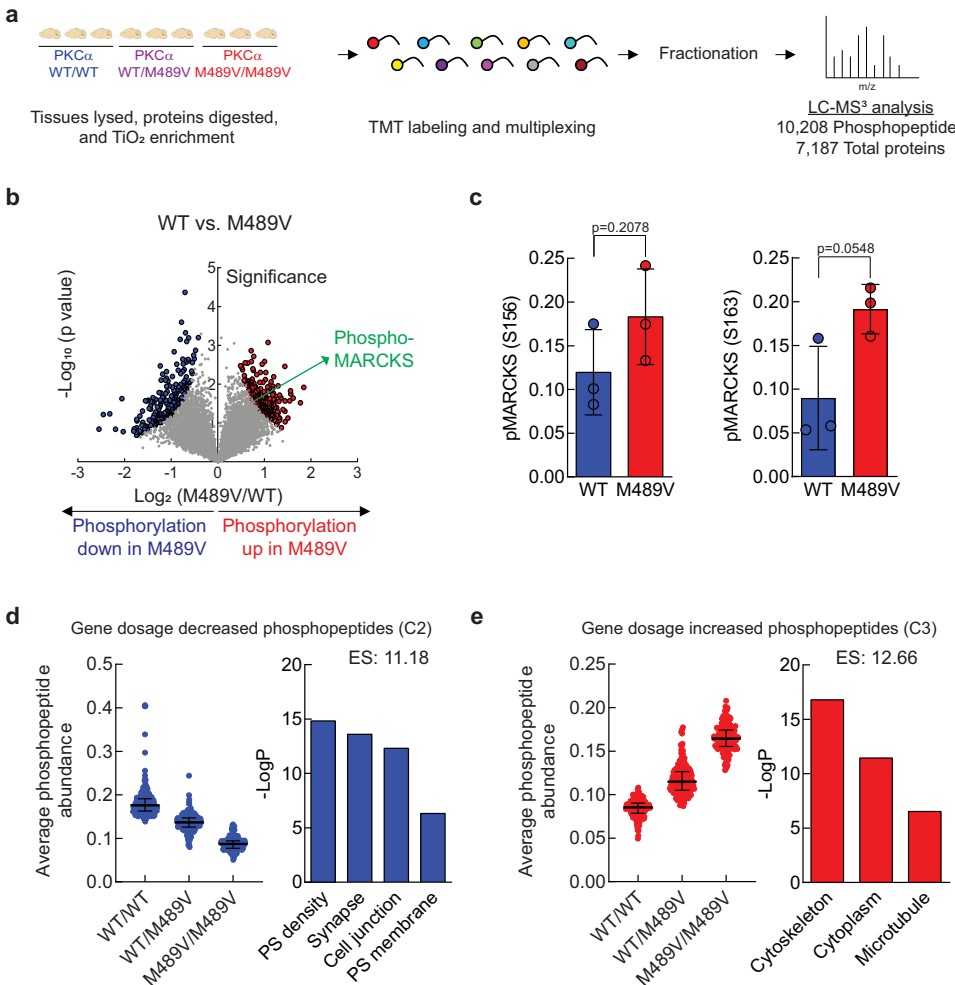

**Fig. 1 | Phosphoproteomics analysis of brains from 3-month-old WT mice and mice harboring the PKCα M489V mutation on C57BL/6 background.**
**a** Experimental design. Brains from WT (blue), heterozygous (purple), or homozygous (red) mice were subjected to phosphoproteomics analysis. 7187 proteins and 10,208 phosphopeptides were quantified per sample in the standard proteomics and phosphoproteomics analyses, respectively. **b** Volcano plot of phosphopeptides quantified from brains from WT mice and M489V homozygous mice. The $-\log_{10}$-transformed $p$ values associated with individual phosphopeptides are plotted against the $\log_2$-transformed fold change in abundance between WT and M489V homozygous brains. Color intensities depict peptides whose phosphorylation level is significantly higher (red) or lower (blue) in M489V homozygous mice compared to WT. Significance was determined at a pi score cutoff of $\alpha < 0.05$, and $p$ values for calculating pi scores were generated with a two-tailed Student's $t$ test not corrected for multiple hypotheses. **c** Graph showing the quantification of two

MARCKS phosphopeptides in brains from WT (blue) and M489V homozygous (red) mice indicates the mean ± SD ($p$ values determined using a two-tailed Student's $t$ test). **d** Left: graph showing the distribution of peptides whose phosphorylation significantly decreased (values of $p < 0.05$ using a two-tailed Student's $t$ test) in a gene-dose dependent manner (WT > WT/M489V > M489V/M489V). Graph depicts the median± interquartile range. Right. GO enrichment analysis from these peptides using DAVID. (ES = enrichment score, PS = post-synaptic). **e** Left: graph showing the distribution of peptides whose phosphorylation significantly increased (values of $p < 0.05$ using a two-tailed Student's $t$ test) in a gene-dose dependent manner (WT < WT/M489V < M489V/M489V). Graph depicts the median ± interquartile range. Right. GO enrichment analysis from these peptides using DAVID. Data in **b**–**e** are representative from $n = 3$ biological independent samples for each genotype. Source data for **c**–**e** are provided in the Source Data file.

a gene-dosage-dependent manner (C2) (Fig. 1d, right). However, peptides whose phosphorylation increased as a function of the variant (C3) showed significant enrichment in cytoskeleton, cytoplasm, and microtubule function (Fig. 1e, right). These data are consistent with enhanced PKCα function increasing the phosphorylation of direct substrates such as MARCKS to modulate cytoskeletal function, and indirectly decreasing the phosphorylation of substrates that are key regulators of the synapse, either by enhancing phosphatase activity or inhibiting kinases directed at these substrates.

**Spine density loss in hippocampal neurons of M489V PKCα mice**
Given that post-synaptic proteins were one of the most altered in the phosphoproteomic analysis of M489V brain, we examined whether the synapses in the M489V mice had an altered morphology compared to those from WT mice. The post-synaptic component of excitatory

synapses in the brain is comprised of small extensions on dendrites known as dendritic spines[35,36]. The morphology of the spines is an indicator of the stability, plasticity, and strength of associated synapses. The majority of excitatory synapses in the brain exist on dendritic spines and, accordingly, the regulation of dendritic spine density in the hippocampus is considered to play a central role in learning and memory[37,38]. To better understand the link between enhanced PKC activity and neurodegeneration, we examined spine density in hippocampal neurons from 4.5-month-old littermate male WT mice or mice homozygous for the PKCα M489V variant. We observed a slight (9.81 ± 0.02%), but statistically significant, reduction in the number of spines per micron in the M489V homozygous mice compared with littermate WT mice, as assessed by fluorescence staining of neuronal projections (Fig. 2a, b). Furthermore, western blot analysis of isolated hippocampi revealed a general increase in the phosphorylation of PKC substrates in the PKCα

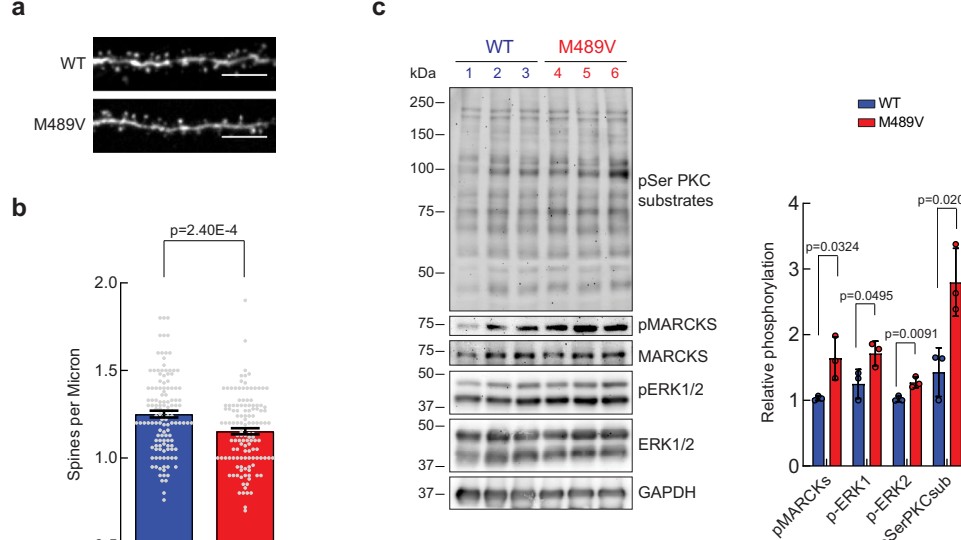

**Fig. 2 | The AD-associated PKCα M489V mutation reduces spine density and increases PKC substrate phosphorylation in the hippocampus. a** Representative immunofluorescence image of dendritic segments from hippocampal neurons injected with Alexa Fluor® 594 Hydrazide to follow neuronal projections (dendritic segment = 20 μm in length, scale bar 5 μm). $n = 129$ dendritic segments from 6 mice of each genotype were analyzed. **b** Average number of spines per micron in neurons isolated from 4–5-month-old WT or littermate M489V PKCα homozygous male mice ($n = 6$ mice of each genotype). Spines were counted in separate spine segments (10–25 μm in length). Data show a spine density reduction of -10%, from 1.25 ± 0.02 spines/μm to 1.15 ± 0.02 spines/μm in the M489V mice. $n = 129$ dendritic segments (over 1500 spines) were analyzed. Error bars show standard error of the mean. ($p = 2.40E-4$ using unpaired two-tailed Student's $t$ test). **c** Left. Immunoblots

of lysates of hippocampi obtained from WT mice (lanes 1–3, blue) or M489V mice (lanes 4–6, red). Right. Relative phosphorylation represents the densitometric analyses of the western blot phosphorylation signal and the total antibody signal of the indicated substrates. pERK1/2 (T202/Y204 for ERK1 and T185/Y187 for ERK2) and pMARCKS (S159/S163) signal was normalized to total ERK1/2 and total MARCKS signal respectively, and phospho-Ser PKC substrates signal was normalized to its GAPDH loading control. Data were normalized to the WT1 values, and normalized data from the depicted western blots were plotted as average normalized intensity ± SEM ($p$ values were determined using a two-tailed Student's $t$ test). Data are representative of $n = 3$ biologically independent experiments. Source data and uncropped blots are provided in the Source Data file.

M489V samples compared to WT (Fig. 2c). The phosphorylation of MARCKS at Ser159/Ser163 (Fig. 2c) was also increased in the PKCα M489V hippocampal samples, consistent with the whole brain phosphoproteomics data (Figs. 1b, c) and previously validated by western blot[34]. Additionally, the phosphorylation of the extracellular signal-regulated kinase (ERK) 1/2, a MAP kinase family member, was enhanced in the PKCα M489V hippocampi compared to those from WT mice (Fig. 2c). In summary, the PKCα M489V variant mice display reduced spine density, as well as enhanced phosphorylation of proteins that regulate neurites, an important hallmark of Alzheimer's disease.

## Impaired cognition of PKCα M489V mice on a C57BL/6 background

Because loss-of-spine density in the hippocampus correlates with decreased learning ability[39], we next assessed whether the PKCα M489V variant impacted cognition. We examined the behavior of WT and M489V mice on the Barnes maze, a test that has been widely used to assess spatial learning and memory in AD[40]. In this test, mice are trained to use distinct cues around the maze to find an escape box under one of 20 holes around the perimeter of a round platform (Fig. 3a). After the training period, the escape tunnel is removed, and the amount of time that mice spend in each quadrant (the target quadrant vs non-target quadrants) searching for the hole is recorded. Cognitively intact mice spend more time in the target quadrant relative to the other areas of the maze, whereas mice with impaired cognition do not discriminate well between the four quadrants. WT mice of all ages performed well on this test, spending approximately twice as long in the target quadrant compared to the other quadrants (Fig. 3b–d). In marked contrast, mice harboring the M489V mutation spent progressively less time in the target quadrant with age, such that at 12 months, they no longer discriminated between the target and

other quadrants (Fig. 3d). Even at 3 months of age, their ability to recognize the target quadrant was decreased relative to the WT mice (34 ± 2% vs 41 ± 5% of time in target quadrant, respectively). This reduced time in the target quadrant was not a reflection of impaired locomotion, as activity levels were the same in WT and M489V mice in both the Barnes maze probe test and an independent test of activity (Supp Fig. 2a and Supp Fig. 2b). Additionally, the M489V mice did not show increased anxiety as assessed by the light/dark test (Supp Fig. 2c). Thus, the M489V mice had impaired learning and memory but not alterations in mobility or anxiety compared with the WT mice. These results establish that the AD-associated mutation M489V in PKCα is sufficient to cause cognitive decline in C57BL/6 mice.

## Enhanced Aβ-induced synaptic depression in neurons of PKCα M489V mice

Alzheimer's disease is characterized by the presence of neuritic plaques, which primarily consist of Aβ peptides derived from APP proteolysis[41,42]. These Aβ aggregates are known to degrade synapses and impair memory formation[31,43]. Because PKCα is necessary for Aβ-induced synaptic depression[21,31,44], we reasoned that the enhanced activity of the PKCα M489V AD mutation might enhance electrophysiological responses to Aβ. Organotypic hippocampal slices from WT or M489V mice were infected with a Sindbis viral vector that expresses CT100, a product of APP cleavage by β-secretase. As a result of CT100 expression, Aβ peptide production increases in infected neurons due to processing by γ-secretase[6,45,46], resulting in synaptic depression[45,47–49]. Synaptic transmission was studied 18–24 h after infection by obtaining whole-cell recordings from two neighboring pyramidal hippocampal CA1 neurons, one infected and one uninfected; electric stimulation of Shaffer collateral axons was used to elicit AMPAR-mediated excitatory post-synaptic currents (EPSCs) (Fig. 4a).

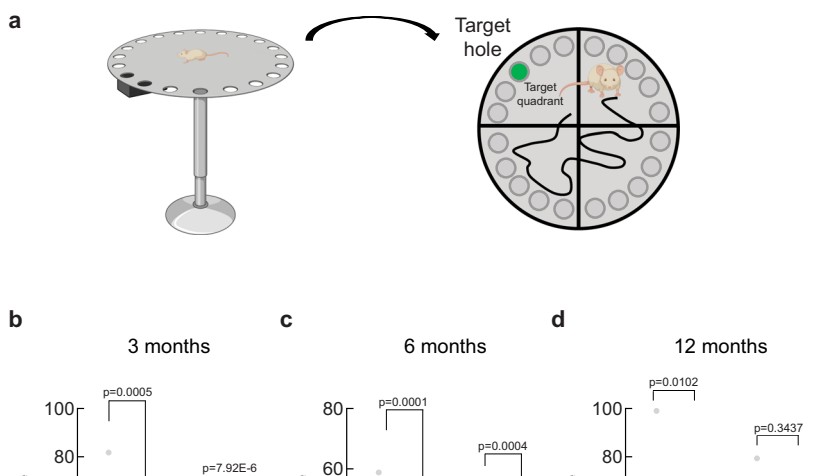

**Fig. 3 | Impaired spatial learning and memory in C57BL/6 PKCα M489V/M489V mice in the Barnes maze test. a** Schematic of the Barnes maze test. Created with Biorender.com. **b**–**d** Percent time spent in the target quadrant (filled bars) versus the average of the other quadrants (open bars) in the probe test in separate groups of 3 (**b**), 6 (**c**), and 12 (**d**) months old mice. Group sizes: 3-month (WT: 4 males, 8 females, M489V littermates: 6 males, 6 females), 6-month (WT: 8 males, 8 females, M489V littermates: 8 males, 8 females), 12 month (WT: 7 males, 12 females, M489V littermates: 12 males, 11 females). Error bars show standard error of the mean. No statistically significant sex differences were found. ANOVA was used for statistical analysis, followed by post hoc two-tailed Student's *t* test to determine the *p* values. Source data are provided in the Source Data file.

Such cell-pair recordings permit one to compare directly the impact of elevated Aβ on synaptic transmission, as the number of activated Shaffer collateral axons targeting infected and non-infected cells is on average the same, irrespective of the stimulation intensity. As expected[48,49], neurons expressing CT100 displayed synaptic depression compared to uninfected neighboring neurons in both WT and M489V slices. However, this effect was more pronounced in the slices from the M489V mice (26 ± 12% depression in WT mice; 55 ± 6% depression in M489V mice; *p* < 0.05; Fig. 4b–d). These results indicate that Aβ-dependent synaptic depression in neurons is enhanced by the presence of a more active PKCα, supporting a model in which increased PKC activity drives the synaptic loss associated with AD.

**Phosphoproteomic profiling of tg-AD mouse with M489V PKCα variant**

The enhanced synaptic depression induced by Aβ in PKCα M489V hippocampal neurons compared to WT neurons led us to next address whether the pathology associated with the gain-of-function PKCα mutation aggravates the effects caused by the presence of the APP. To this end, the M489V mutation was introduced onto a B6;SJL mouse with the *APP* transgene carrying the Swedish mutation (tg-AD); this well-established mouse model has a predisposition to AD as a result of elevated Aβ levels caused by abnormal processing of the mutant APP by β-secretase[50]. We first investigated whether the introduction of the PKCα M489V mutation into the tg-AD mouse model with the APPswe transgene impacted the presence of soluble and insoluble Aβ in the brain. Brains were homogenized and different fractions of Aβ were extracted and analyzed by enzyme-linked immunosorbent assay (ELISA). The levels of Aβ–40 and Aβ–42, as well as the Aβ–42/Aβ–40 ratio, were the same in mice with WT PKCα or the M489V variant (Supp Fig. 3a–c). These results were confirmed by histochemical analyses, where brains both WT tg-AD and M489V tg-AD mice presented similar levels of Aβ plaques (Supp Fig. 3d). This suggests that the molecular events by which enhanced

PKCα activity leads to neurodegeneration and cognitive decline are downstream or independent of Aβ-production.

Given that the PKCα M489V variant did not influence Aβ-production in the tg-AD mouse model, we reasoned it is signaling downstream of Aβ and could further deregulate the phosphoproteome, on top of the changes induced by the *APPswe* transgene. We thus examined whether the PKCα M489V altered the phosphoproteome in the AD mouse model in a similar fashion as the alterations observed on the non tg-AD background. Brains from mice harboring WT or M489V PKCα in the presence (WT tg-AD/M489V tg-AD) or absence (WT non tg-AD/M489V non tg-AD) of the *APPswe* transgene were isolated at 4.5 and 6 months of age. Tissues were processed and analyzed as in Fig. 1a to quantify the proteome and phosphoproteome (Fig. 5a). We quantified ~12,000 phosphopeptides per sample, representing ~6500 proteins with an FDR of <1% at the peptide and protein level (Dataset S2). The median within sample coefficient of variation was, on average, <20%, indicating minimal variance (Supp Fig. 4a). Consistent with previous experiments, the phosphosite distribution was 77.41% phosphoserine (pS), 19.40% phosphothreonine (pT), and 3.19% phosphotyrosine (pY) (Supp Fig. 4b), with most of the peptides phosphorylated on single (73.60%) and double (23.10%) sites, and fewer on triple (2.96%) and quadruple (0.35%) sites (Supp Fig. 4c).

We next analyzed the APP and PKCα levels in the different cohorts of mice. As expected, the amount of APP was consistently elevated in all the tg-AD mice (WT tg-AD/M489V tg-AD). Notably, the relative abundance of PKCα was the same for every group (Fig. 5b). These data highlight the reproducibility of this analysis and validate our biochemical studies[34] showing that the M489V mutation does not alter the stability/steady-state levels of PKCα in vivo. Western blot analysis of whole brain lysate revealed that the APP transgene increased PKC activity, as observed by increased phosphorylation of PKC substrates in the brains of mice carrying the APP transgene compared with the control mice (Fig. 5c). This is consistent with the phosphoproteomic

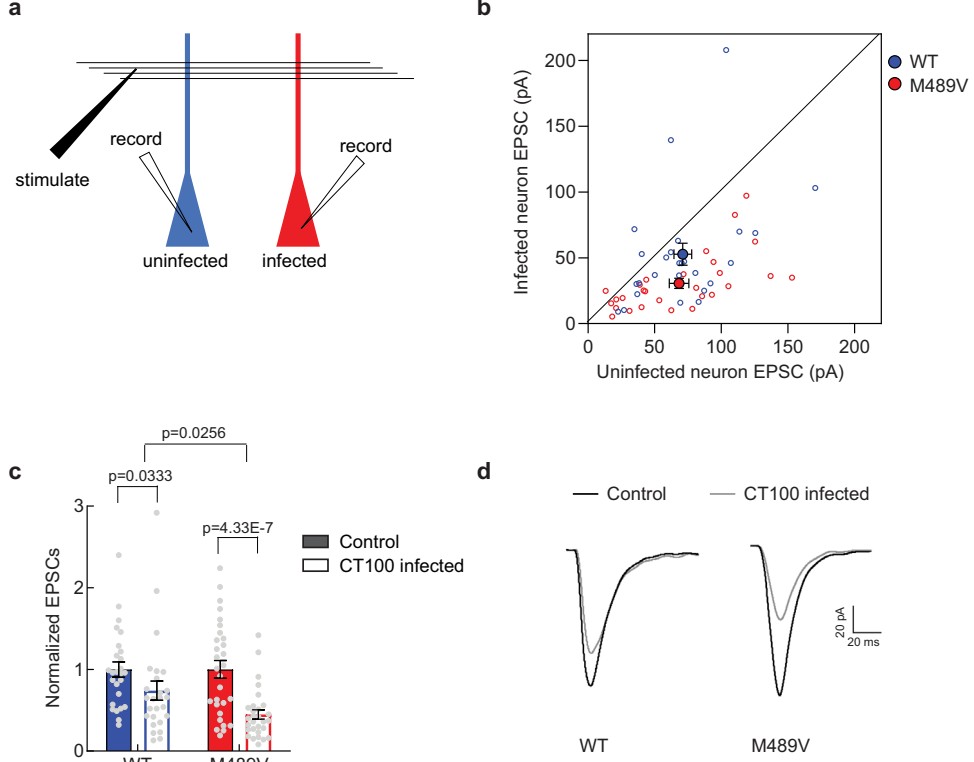

**Fig. 4 | The AD-associated PKCα M489V mutation exacerbates Aβ-induced synaptic depression. a** Experimental design of dual-patch whole-cell recordings (see Material and Methods section). **b** Dot plot showing the individual excitatory post-synaptic currents (EPSCs) in CT100 infected versus uninfected CA1 hippocampal neurons from WT (blue open circles, $N = 26$) and M489V (red open circles, $N = 29$) mice. Group averages indicated by filled circles, error bars indicate SEM. **c** Bar graph of dual-patch recordings for indicated groups (same data as in **b**, $n = 26$ wt and $n = 29$ M489V CA1 hippocampal neurons); responses were normalized to

uninfected controls (solid bars) and graph depicts the mean ± SEM. Paired two-tailed $t$ tests were used to assess statistical significance of dual-patch recordings and obtain $p$ values. To compare CT100-induced depression in WT versus M489V mice, two-way ANOVA was performed ($p < 0.0001$), followed with Newman–Keuls multiple comparison post hoc test to determine the $p$ value. **d** Representative traces (mean of at least 40 consecutive trials) obtained from evoked AMPA-receptor-mediated responses from WT and M489V CA1 hippocampal neurons. Source data are provided in the Source Data file.

analysis which revealed increases in a large number of substrates (top 25 phosphosites enhanced by PKC-M489V in the tg-AD mice are presented in Supp. Tables 2 and 3).

Analysis of the changes in the phosphoproteome resulting from the introduction of the PKCα M489V mutation in an AD mouse model, using k-means clustering of the 28,084 phosphopeptides, revealed nine distinct clusters (Supp Fig. 4d). The age-related phosphoproteome clustered into three distinct groups: C1, C6, and C8. C1 and C8 contained peptides whose phosphorylation increased with age, while C6 contained peptides that displayed a modest decrease in phosphorylation with age. M489V caused an increase in phosphorylation only at the early age group (4.5 months) in the C2 and C4 clusters of the APP brain phosphoproteome. The C3 and C5 clusters contained peptides whose phosphorylation decreased at 4.5 months. C9 was of particular interest (Supp Fig. 4d) as it contained proteins whose phosphorylation increased only slightly from 4.5 months to 6 months of age in tg-AD mice with WT PKCα, but increased substantially with age in mice harboring the PKCα M489V variant (Fig. 5d, left). We reasoned that this set comprised proteins whose phosphorylation increased with age in the tg-AD background in a manner that was exacerbated with the PKCα mutation. To explore the biological functions associated with this distinct phosphoproteome, we analyzed the biological processes that were overrepresented in C9 using GO analysis. Learning, axonogenesis, and synaptic vesicle endocytosis were significantly enriched in C9, as well as protein phosphorylation, and dendrite development (Fig. 5d, right). In addition, String analysis revealed that many of these proteins were part of the mTOR signaling

pathway, the mitogen-activated protein kinase (MAPK) signaling pathway and involved in neuron projection (Fig. 5e); note that neuron projection processes were also significantly perturbed by the M489V variant on a WT background (e.g., Fig. 1d, e). To further characterize the impact of the PKCα mutation on these pathways, the phosphorylation of ERK1/2 was examined. It has been previously described that APP expression and exposure to oligomeric Aβ peptides enhance Ras/ERK signaling, increasing anomalous proliferation and subsequent neurodegeneration[51,52]. Immunoblot analysis revealed increased phosphorylation of ERK1/2 in brain lysates from both the M489V non-tg-mice and the tg-AD mice compared to WT non tg-AD mice (Fig. 5f). This increase was also captured by the phosphoproteomics analysis (Fig. 5g). Thus, both the PKCα M489V mutation or the APP transgene enhance Erk signaling. Overall, phosphoproteomics analysis indicates that a slight enhancement in the catalytic activity of PKCα induces changes in protein phosphorylation in the brains of mice carrying the APP transgene at both 4.5 and 6 months of age. This altered signaling could lead to cell cycle deregulation, aberrant proliferative signaling and subsequent memory and learning disability at older ages.

## Acceleration of impaired cognition in tg-AD mouse with PKCα M489V

The above studies indicate that the presence of the PKCα M489V AD variant is sufficient to impact the brain phosphoproteome at 3 months and learning ability in C57BL/6 mice at 12 months of age. To elucidate whether the presence of APP would aggravate or accelerate this effect on cognition, spatial learning and memory was assessed using the

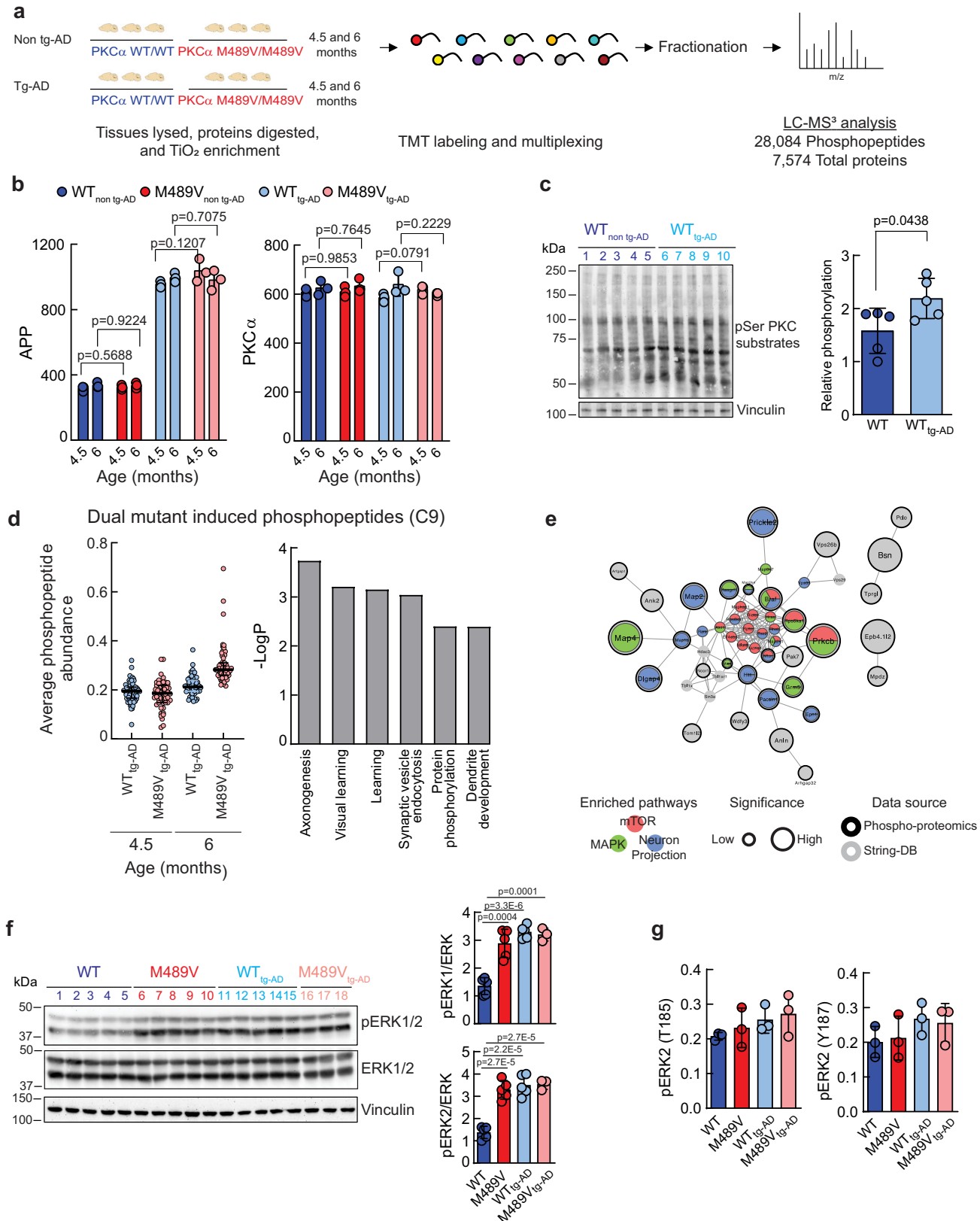

Barnes maze test in mice harboring the PKCα M489V mutation (red) and WT littermate controls (blue) either without (WT and M489V) or with (WT tg-AD/M489V tg-AD) the *APP_swe* transgene at 4.5 (Fig. 6a), 6 (Fig. 6b) and 12 months of age (Fig. 6c). Introduction of this PKCα mutation onto the B6;SJL background without the APP transgene (non tg-AD) did not cause cognitive impairment at either of the early ages

tested (Figs. 6a, b), but deficits were apparent at 12 months of age (Fig. 6c). Thus, the PKCα mutation alone was sufficient to cause cognitive impairment in the non tg-AD mice, as reported for the C57BL/6 mice (Fig. 3d). The presence of the *APP_swe* transgene did not result in deficits in Barnes maze performance at the youngest age of 4.5 months in either WT tg-AD/M489V tg-AD mice (Fig. 6a), but it did produce

**Fig. 5 | Phosphoproteomics analysis of brains from WT mice and mice harboring the PKCα M489V mutation (red) on a B6;SJL background with the APP transgene carrying the Swedish mutation (APP_swe). a** Experimental design. Brains from WT (blue) or homozygous (red) mice with or without the APP_swe transgene at 4.5 and 6 months of age were subjected to phosphoproteomics analysis. 7574 proteins and 28,084 phosphopeptides were quantified in the standard proteomics and phosphoproteomics analyses, respectively. **b** Graphs showing the abundance of APP (left) and PKCα (right) detected in the proteomic analysis across all samples. Graphs depicts the mean ± SD (*p* values obtained using a two-tailed Student's *t* test, *n* = 3 biologically independent samples). **c** Left. Immunoblot of brain lysates obtained from WT non tg-AD mice (lanes 1–5, dark blue) and tg-AD mice (lanes 6–10, light blue). Right. Relative phosphorylation represents the quantification of PKC substrates phosphor-signal normalized to vinculin. Normalized data from the depicted western blot were plotted as average normalized intensity ± SEM (*p* = 0.0438, using a two-tailed Student's *t* test). **d** Left: graph

showing the distribution of phosphopeptides from C9 Graph depicts the median± interquartile range (values of *p* < 0.05 using a two-tailed Student's *t* test, *n* = 3 biological independent samples). Right. GO enrichment analysis of C9 peptides using DAVID. **e** STRING analysis of proteins whose phosphopeptides were present in C9. Red represents proteins in the mTOR signaling pathway, green in the MAPK signaling pathway and blue represents proteins involved in neuron projection. **f** Left. Immunoblot of pERK1/2 (T202/Y204 for ERK1 and T185/Y187 for ERK2) and total ERK1/2 in brain lysates obtained from WT non-tg-AD (WT), M489V non tg-AD (M489V), WT tg-AD and M489V tg-AD mice at 6 months. Right. ERK1 (top) or ERK2 (bottom) phosphorylation reflects the normalized phospho-signal relative to total ERK. Data represent the average normalized intensity ± SEM (ANOVA was used for statistical analysis, followed by post hoc two-tailed Student's *t* test). **g** Quantification of ERK phosphopeptides detected by phosphoproteomics in 6-month-old samples (mean ± SD). Source data and uncropped blots for **c** and **f** are in the Source Data file.

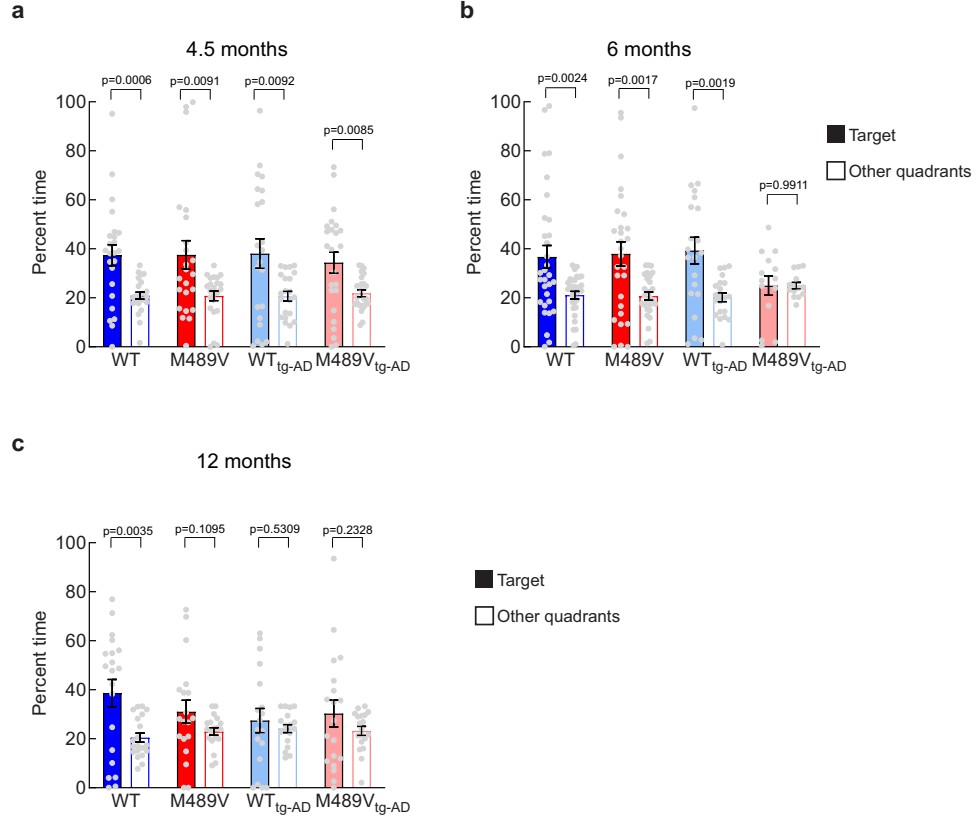

**Fig. 6 | Impaired spatial learning in the Barnes maze test in PKCα M489V mice + AD transgenic mouse.** Percent time in the target quadrant (filled bars) vs. the average of the other quadrants (clear bars) in the probe test in separate groups of 4.5 (**a**), 6 (**b**) and 12 (**c**) month-old mice. Group sizes: 4.5-month (WT non tg-AD (WT): 12 males, 12 females, M489V non tg-AD (M489V): 11 males, 12 females, WT harboring the APP_swe transgene (WT tg-AD): 11 males, 11 females, M489V harboring the APP_swe transgene (M489V tg-AD): 12 males, 12 females), 6-month (WT: 15 males,

16 females, M489V: 12 males, 16 females, WT_APP: 10 males, 11 females, M489V_APP: 8 males, 6 females), 12 month (WT: 10 males, 10 females, M489V: 10 males, 9 females, WT_APP: 9 males, 9 females, M489V_APP: 5 males, 14 females). Error bars show standard error of the mean. No statistically significant sex differences were found. ANOVA was used for statistical analysis, followed by post hoc two-tailed Student's *t* test to determine the *p* values. Source data are provided in the Source Data file.

cognitive deficit at 12 months (Fig. 6c). Strikingly at 6 months, the tg-AD mice with WT PKCα (WT tg-AD) had normal cognition, but the presence of the M489V mutation (M489V tg-AD) abolished the ability of the mice to discriminate between the target quadrant (filled bars) and the other quadrants (open bars) (Fig. 6b). Thus, the PKCα M489V mutation accelerated the cognitive decline in the AD mouse model. It is important to highlight that neither M489V or M489V tg-AD mice showed reduced activity levels during the Barnes maze probe test or locomotor activity test, nor increased anxiety-like behavior in the light/dark test (Supp Fig. 5). Thus, this AD-associated mutation in PKCα, which enhances the catalytic activity of the enzyme by 30%[34],

dramatically impaired cognition of mice at 6 months of age when paired with the *APP_swe* transgene (Fig. 6b) and was sufficient to affect learning and memory on its own at 12 months of age (Fig. 3d and Fig. 6c). These findings establish that 1] the M489V mutation in PKCα is sufficient, alone, to cause a behavioral defect in mice, and 2] the cognitive decline associated with *APP_swe* in this mouse line is accelerated in mice also harboring the PKCα mutation.

### Increased PKCα levels in human Alzheimer's disease brain

The above data indicate that enhanced PKCα signaling resulting from a rare, but highly penetrant variant of PKCα is sufficient to

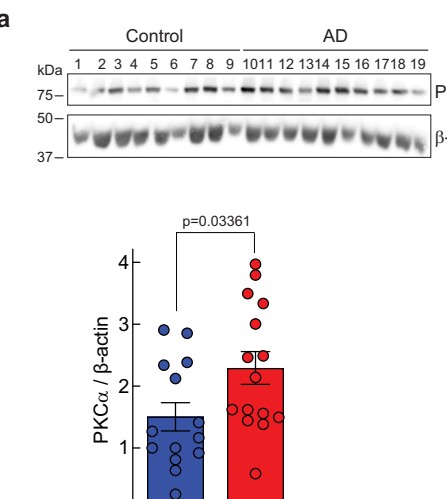
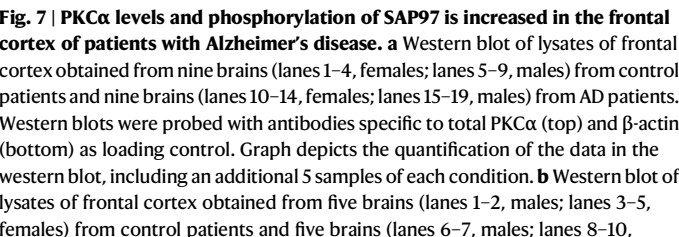
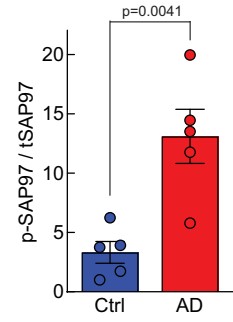

**Fig. 7 | PKCα levels and phosphorylation of SAP97 is increased in the frontal cortex of patients with Alzheimer's disease. a** Western blot of lysates of frontal cortex obtained from nine brains (lanes 1–4, females; lanes 5–9, males) from control patients and nine brains (lanes 10–14, females; lanes 15–19, males) from AD patients. Western blots were probed with antibodies specific to total PKCα (top) and β-actin (bottom) as loading control. Graph depicts the quantification of the data in the western blot, including an additional 5 samples of each condition. **b** Western blot of lysates of frontal cortex obtained from five brains (lanes 1–2, males; lanes 3–5, females) from control patients and five brains (lanes 6–7, males; lanes 8–10,

females) from AD patients. Western blots were probed with antibodies specific to a known PKC phosphorylation site on SAP97 (Thr656) (Top) or to total SAP97 (Bottom). Graph depicts the quantification of the data in the western blot. **a, b** Bands were quantified using densitometry and the phospho-SAP97 signal was normalized to total SAP97 signal and PKCα signal was normalized to its β-actin loading control. Normalized data from the depicted western blots were plotted (Bottom) as average normalized intensity ± SEM (*p* values were determined using unpaired two-tailed Student's *t* test). Source data and uncropped blots for figures **a** and **b** are in the Source Data file.

---

impair cognition. Given that elevated PKC signaling has been detected as one of the earliest events in the pathology of AD[29], we asked whether the steady-state levels of PKCα might be elevated in AD, resulting in enhanced signaling output. In this regard, the amount of PKC protein in cells regulates its signaling output[22], with higher steady-state levels resulting in higher signaling output; for example, the related isozyme, PKCβII, has been shown to be haplo-insufficient in suppressing oncogenic signaling[53]. Thus, we assessed the levels of PKCα protein in the frontal cortex of human brains from deceased patients with AD or control individuals by immunoblot as a measure of whether enhanced PKCα signaling output is associated with AD. This analysis revealed a statistically significant 20% increase in the steady-state levels of PKCα in the post-mortem brain of AD patients (Fig. 7a). Additionally, as another marker for PKC activation, we assessed the phosphorylation state of T656 on SAP97, a previously identified PKCα phosphorylation site[32]. Phosphorylation of T656 was increased by approximately four-fold in the brains from AD patients compared to control patients, consistent with higher PKC activity (Fig. 7b). These data reveal that PKCα is generally upregulated in human AD brain, resulting in enhanced substrate phosphorylation.

## Discussion

Here we show that a highly penetrant, AD-associated mutation in PKCα is sufficient to cause impaired cognition in a mouse model. Furthermore, this mutation has a synergistic effect with enhanced APP production to exacerbate cognitive decline in the tg-AD mouse model. Thus, this mutation is not only sufficient to cause pathologies associated with AD, but likely does so by pathways separate from APP-induced AD. This mutant PKCα (M489V), which exhibits a 30% increase in catalytic rate, rewires the brain phosphoproteome, reduces the spine density in hippocampal neurons, enhances Aβ-dependent synaptic depression, and ultimately impairs cognition without impacting Aβ levels in the brain. Our work identifies PKCα inhibition as

a therapeutic strategy in AD which may be generally relevant as elevated PKCα protein, and hence signaling output, is associated with human AD patients.

Although PKC was originally discovered in the brain over four decades ago[54,55], it has only recently been recognized as an emerging biomarker and therapeutic target in neurodegeneration. Specifically, the advent of unbiased phosphoproteomic approaches and whole-genome sequencing puts the spotlight on PKCα, with mounting data pointing to a role in neurodegenerative diseases such as Alzheimer's. PKC isozymes are involved in numerous brain disorders such as glioblastoma, cerebral ischemia, addiction, and neurodegeneration[56–59], and their well-characterized effects on Aβ-mediated synaptic depression and phosphorylation of tau have been proposed mechanisms that lead to synaptic damage, neurotoxicity, and cognitive impairment[3,4,59]. However, it is the unbiased approaches that provide the most compelling evidence for the critical role of PKC in maintaining normal brain function. A comprehensive phosphoproteome analysis by Tagawa et al.[29] revealed that PKC substrates account for over half of the core molecules that display increased phosphorylation in AD brains, a finding supported by subsequent phosphoproteomics studies identifying PKC as one of the main kinases activated in AD[30], and reporting increased phosphorylation of PKCα at Thr638[60], a quantitatively phosphorylated C-terminal site that serves as an indicator of PKC steady-state levels[61,62]. Gain-of-function variants in PKCα segregating with affected family members in LOAD first drew specific attention to this conventional PKC isozyme[21]. In this study, we have leveraged one of these AD-associated rare variants, PKCα-M489V, to establish that a small increase in catalytic rate of PKCα is sufficient to drive cognitive decline in a mouse model. This particular variant was ideally suited for our study because the Met to Val substitution has no effect on auto-inhibition, on/off dynamics, or protein stability of PKCα, as established by biochemical and cellular analyses. Instead, the variant catalyzes 7 reactions per second, rather than the 5 reactions per second catalyzed by WT PKCα when in the active conformation[34]. When introduced into a mouse model by genome editing, the steady-state levels of the

variant M489V PKCα are the same as that of the WT enzyme, validating the biochemical analysis that the mutation does not alter the stability of PKCα. Thus, this mutation provided an ideal model to interrogate whether a small enhancement in signaling output of PKCα is sufficient to cause cognitive decline. Strikingly, this small change in activity was sufficient to alter the brain phosphoproteome, reduce the spine density in hippocampal neurons, and impair cognition. Furthermore, analysis of post-mortem human brain revealed that, specific mutations aside, patients with AD generally have higher steady-state levels of PKCα. The steady-state levels of PKC are precisely set by diverse mechanisms, including stabilizing phosphorylations mediated by mTORC1 and PDK-1 and opposed by the quality control phosphatase PHLPP, as well as regulators such as Pin1, E3 ligases, and heat shock proteins[63]. This suggests that not only are mutations in PKCα biomarkers for the disease, but the intrinsic set point of PKCα levels, controlled by diverse regulators, may also predict susceptibility to AD.

Gene Ontology analysis of the proteins whose phosphorylation increased in an M489V gene-dosage dependent manner (i.e. WT/WT < WT/M489V < M489V/M489V) identified components involved in the post-synaptic density and synapse processes. Proteins whose phosphorylation increased with increasing PKCα activity are likely downstream substrates of PKC whose phosphorylation modulates synaptic structure and function. PKC has previously been reported to phosphorylate numerous substrates to promote long term depression[64,65], including the dopamine transporter[66] and glutamate receptors such as α-amino-3-hydroxy-5-methyl-5-isoxazolepropionic acid (AMPA)-type glutamate receptors (AMPARs) and N-Methyl-D-aspartic acid or N-Methyl-D-aspartate (NMDA)-type glutamate receptors (NMDARs)[64,65,67]. Conversely, there was a group of peptides whose phosphorylation decreased in a gene-dosage-dependent manner. These are likely indirectly regulated downstream of PKC, either as a result of enhanced phosphatase activity of inhibited kinase activity directed at these substrates. Proteins in this group were involved in cytoskeleton and microtubule processes, supporting numerous studies on PKC regulating diverse cellular processes that culminate in cytoskeleton regulation. For example, phosphorylation of MARCKS and GAP43 by PKC promotes their translocation from the plasma membrane into the cytosol, triggering the depolarization and disruption of actin filaments[68,69]. Additionally, PKC regulates tau phosphorylation, a microtubule-associated protein, and subsequently modulates the cytoskeleton dynamics in neurons[70–72]. Taken together, increased PKC activity in the M489V mice may be disrupting the dynamics of microtubules and thus affecting the maintenance and formation of synapses in the brain.

Consistent with aberrations in the maintenance and formation of synapses, the M489V mice displayed a small but highly significant reduction in spine density in hippocampal neurons. PKC-catalyzed phosphorylation of MARCKS is well established to regulate dendritic spine morphology, with increased phosphorylation associated with reduced spine density[73,74]. Furthermore, MARCKS phosphorylation has been proposed as a marker for degeneration, with elevated MARCKS phosphorylation preceding pathologies such as Aβ aggregation in several mouse models of AD[29]. In our phosphoproteomic analysis of brains from WT and M489V mice, we observed an increase in MARCKS phosphorylation at Ser159/Ser163, consistent with previous studies that indicate that brains from mice harboring the M489V mutation in PKCα exhibit higher MARCKS phosphorylation[34]. Taken together, the increased MARCKS phosphorylation resulting from the PKCα M489V variant may drive the neurite degeneration observed in the M489V mice. In correspondence with neurite degeneration, several studies have also correlated synapse density loss with memory deficits[37,38]. Our behavioral studies support a correlation between spine density loss with neurodegeneration. Specifically, Barnes maze tests revealed that the M489V variant of PKCα alone, without the presence of APP, was able to induce cognitive impairment. These data indicate that enhanced PKCα alone is sufficient to cause neurodegeneration. The finding that PKCα alone is able to cause impaired learning, with no impact of the variant on Aβ levels, suggests that PKCα may lead to neurodegeneration in an APP-independent manner. Alternatively, PKCα may act downstream of APP, accounting for why the variant accelerates APP-induced cognitive decline.

The molecular mechanisms by which enhanced PKCα activity leads to a cognitive deficit in a mouse model await elucidation. One clue to the puzzle is that PKCα transduces signaling downstream of Aβ. Previous studies have shown that Aβ fails to induce synaptic depression in hippocampal slices from PKCα knock-out animals[21]. Here we show that, conversely, Aβ-induced synaptic depression is exacerbated in animals with the activity-enhancing M489V variant. Furthermore, pharmacological and genetic approaches suggest that the synaptic depression effects of Aβ are mediated by PKCα acting on a protein scaffold via its PDZ ligand[21]. These results suggest that enhanced PKCα activity causes synaptic depression not only by increasing the phosphorylation and subsequent internalization of known membrane substrates such as GluR2[65], but also through the phosphorylation of its interaction partners such as SAP97[32]. The regulation of PDZ domain proteins by PKC may be an important mechanism to consider in the depressive effects of Aβ on synapses.

Our findings underscore the remarkable success of GWAS analysis in identifying functional AD variants that are tractable targets for therapy and can serve as biomarkers for the disease. As biomarkers, mutations in PKCα serve as a powerful diagnostic for disease susceptibility in AD, in the same way BRCA mutations are used as a diagnostic tool for breast cancer[75]. Additionally, the steady-state levels of PKCα may serve as a diagnostic for disease susceptibility, as our analysis of post-mortem brains revealed that AD patients generally have higher steady-state levels of this enzyme. As a target, it is noteworthy that pharmacological inhibitors or aprinocarsen, a PKCα antisense oligonucleotide, which failed in clinical trials for cancer, could be repurposed for AD[28]. Indeed, the use of specific PKC antisense oligonucleotides to reduce PKC levels is an attractive potential treatment of neurodegenerative diseases, as antisense strategies have been shown to successfully reduce LRRK2 protein levels in Parkinson's disease treatment[76], reduce superoxide dismutase 1 in amyotrophic lateral sclerosis[77], and improve clinical symptoms of patients with spinal muscular atrophy[78–80]. It should be noted that therapeutic strategies would only need to modestly reduce PKC activity, tuning activity down to homeostatic levels. The druggability of kinases, coupled to our detailed understanding of the molecular mechanisms of PKC, poise PKCα as an attractive target in AD.

## Methods

All procedures involving animals were approved by The Scripps Research Institute's Institutional Animal Care and Usage Committee (IACUC) (Protocol number 09-0004) and the University of California San Diego IACUC (Protocol numbers S06288 and S11286), and met the guidelines of the National Institute of Health.

### Mice

**PKCα-M489V mouse generation.** C57BL/6NTac-*Prkca* mice containing the M489V mutation in *Prkca* were generated by Taconic Biosciences GmbH for Cure Alzheimer's Fund as previously described (Supp Fig. 1a)[34].

***APPswe + PKCα-M489V mouse generation.*** This mouse was generated by Taconic Biosciences GmbH for Cure Alzheimer's Fund by an initial intercross of C57BL/6NTac-*PrkcaM489V* homozygous mice and the transgenic *APP_swe* mice (B6;SJL-Tg(APPSWE)2576Kha, model 1349, Taconic)[50]. From this intercross, *PrkcaM489V* heterozygous mice carrying the transgenic *APP_swe* mice (HET;APP) and *PrkcaM489V* heterozygous mice not carrying the transgenic *APP_swe* (HET;NON-APP) were

obtained. Then, the offspring (HET;APP x HET;NON-APP) was inter-crossed to generate the first cohort. Afterwards, HET;APP x HET;NON-APP breeding pairs were maintained to generate all the cohorts used for the experimental studies.

**Housing conditions for the mice.** Mice were group housed in a reverse light cycle room (lights off 8:00AM, on 8:00PM) with behavioral testing occurring in the early dark cycle (active phase), i.e. between 9:00AM and 1:00PM. The housing room was temperature (68–72 F) and humidity (44–61%) controlled.

All procedures involving animals were approved by The Scripps Research Institute's Institutional Animal Care and Usage Committee (IACUC), and the University of California San Diego IACUC, and met the guidelines of the National Institute of Health detailed in the Guide for the Care and Use of Laboratory Animals[81]. The protocol numbers approved for these procedures are 09-0004, S06288, and S11286.

## Behavioral tests

**Barnes maze test.** This is a spatial memory test[82–84] sensitive to impaired hippocampal function[85]. Mice learn to find an escape tunnel among 20 possibilities below an elevated, brightly lit and noisy platform using cues placed around the room. Spatial learning and memory are assessed across trials and then directly analyzed on the final probe trial in which the tunnel is removed and the time spent in each quadrant is determined; the percent time spent in the target quadrant (the one originally containing the escape box) is compared with the average percent time in the other three quadrants. This is a direct test of spatial memory as there is no potential for local cues to be used in the mouse's behavioral decision.

**Locomotor activity test.** Locomotor activity was measured in polycarbonate cages (42 × 22 × 20 cm) placed into frames (25.5 × 47 cm) mounted with two levels of photocell beams at 2 and 7 cm above the bottom of the cage (San Diego Instruments, San Diego, CA). These two sets of beams allowed for the recording of both horizontal (locomotion) and vertical (rearing) behavior. A thin layer of bedding material was applied to the bottom of the cage. Mice were tested for 120 min and data were collected in 5-minute intervals.

**Light/dark test.** The light/dark transfer procedure has been used to assess anxiety-like behavior in mice by capitalizing on the conflict between exploration of a novel environment and the avoidance of a brightly lit open field[86]. The apparatus is a rectangular box made of Plexiglas divided by a partition into two environments. One compartment (14.5 × 27 × 26.5 cm) is dark (8–16 lux) and the other compartment (28.5 × 27 × 26.5 cm) is highly illuminated (400–600 lux) by a 60 W light source located above it. The compartments are connected by an opening (7.5 × 7.5 cm) located at floor level in the center of the partition. The time spent in the light compartment is used as a predictor of anxiety-like behavior, i.e., a greater amount of time in the light compartment is indicative of decreased anxiety-like behavior. Mice were placed in the dark compartment to start the 5-minute test.

ANOVA was used for the statistical analyses of behavioral results, followed by post hoc Student's $t$ tests as appropriate to calculate the $p$ values.

## Electrophysiology

**Organotypic slice cultures.** Organotypic hippocampal slices were prepared from P5-P7 mice pups as previously described[87]. Slice cultures were maintained by changing media every two days, and 18–24 h prior to electrophysiological experiments, slices were infected with Sindbis viruses to express the APP derived peptides CT84 and CT100 as previously described[48].

**Electrophysiological recordings.** Hippocampal organotypic slices were used for electrophysiological recordings shown in Fig. 7. Slices made from PKCα WT and M489V littermates were interleaved. Simultaneous whole-cell recordings were obtained from two neurons, one infected and one neighboring control CA1 pyramidal neurons under visual guidance using differential interference contrast and fluorescence microscopy. One stimulating electrode (contact Pt/Ir cluster electrodes (Frederick Haer)) was placed between 100 and 300 μm down the apical dendrite. Whole-cell recordings were obtained with Axopatch-1D amplifiers (Molecular Devices) using 3–5 MΩ pipettes with an internal solution containing 115 mM cesium methanesulfonate, 20 mM CsCl, 10 mM HEPES, 2.5 mM MgCl$_2$, 4 mM Na$_2$ATP, 0.4 mM Na$_3$GTP, 10 mM sodium phosphocreatine (Sigma), and 0.6 mM EGTA (Amresco), at pH 7.25. External perfusion consisted of artificial cerebrospinal fluid containing 119 mM NaCl, 2.5 mM KCl, 4 mM CaCl$_2$, 4 mM MgCl$_2$, 26 mM NaHCO$_3$, 1 mM NaH$_2$PO$_4$, 11 mM glucose, 0.004 mM 2-chloroadenosine (Sigma), and 0.1 mM picrotoxin (Sigma) (pH 7.4), and gassed with 5% CO$_2$/95% O$_2$ at 27 °C. The AMPAR-mediated excitatory post-synaptic current (EPSC) was measured as peak inward current at a holding potential of −60 mV. Evoked responses were analyzed by averaging 30–100 sweeps using Igor Pro 4.04 software, blind to experimental conditions.

## Mass spectrometry–phosphoproteomics

**Murine brain tissue lysis.** 3-month-old WT and homozygous M489V mice, and 4.5 and 6-month-old WT and homozygous M489V mice with or without the APP transgene were sacrificed and hemibrains were obtained and immediately snap-frozen. Frozen tissue was thawed on ice and homogenized via bead beating in a buffer containing 3% sodium dodecyl sulfate (SDS), 75 mM NaCl, 1 mM NaF, 1 mM β-glycerophosphate, 1 mM Na$_3$VO$_4$, 1 mM sodium pyrophosphate, 1 mM phenylmethylsulfonyl fluoride (PMSF), 1× complete EDTA-free protease inhibitor cocktail from Roche (Basel, Switzerland) and 50 mM HEPES, pH 8.5[88]. Tissues were sonicated with a probe sonicator to ensure full lysis, insoluble cellular debris was removed via centrifugation (16,000×$g$, 10 min, 4 °C), and resultant supernatants were used for downstream processing.

**Protein digestion.** Proteins were denatured by addition of urea (4 M final concentration) then reduced with dithiothreitol (DTT) and alkylated with iodoacetamide (IAA)[89]. Proteins were then precipitated with methanol/chloroform as previously described[89] and dried on a heat block at 56 °C. Dried protein pellets were resolubilized in 1 M urea in 50 mM HEPES, pH 8.5 for digestion in a two-step process (LysC for 16 h at room temperature (RT) followed by Trypsin for 6 h at 37 °C). Digests were acidified by addition of trifluoroacetic acid (TFA), and digested peptides were desalted with C18 Sep-Paks[90]. Desalted peptides were dried, resuspended in 50% acetonitrile/5% formic acid and quantified using the Pierce™ Quantitative Colorimetric Peptide Assay. Peptides from matched samples were aliquoted for both standard proteomics (50 μg) and phosphoproteomics (4 mg) and lyophilized.

**Phosphopeptide enrichment.** Phosphopeptides were enriched by TiO$_2$ beads as previously described[91,92]. Peptides were resuspended in binding buffer (2 M lactic acid, 50% acetonitrile) and incubated with TiO$_2$ beads that were pre-washed 1× with binding buffer, 1× with elution buffer (50 mM KH$_2$PO$_4$, pH 10) and 2× with binding buffer. Enrichment was conducted at a ratio of 1:4 (peptides:beads) for 1 h at RT. Peptide:bead complexes were washed 3× with binding buffer and 3× with wash buffer (50% acetonitrile/0.1% trifluoracetic acid) to remove non-specific binding. Phosphopeptides were then eluted from the beads using 2 × 5 min incubations in elution buffer while vortexing at RT. Enriched phosphopeptides were desalted and lyophilized prior to TMT labeling.

**Tandem mass tag (TMT) labeling.** For both standard and phospho-proteomics, peptides were labeled for quantitation using TMT 10-plex reagents[93,94]. TMT reagents were resuspended in dry acetonitrile to a concentration of 20 μg/μl. Lyophilized peptides were resuspended in 30% acetonitrile in 200 mM HEPES, pH 8.5 and mixed with 8 μl of the appropriate TMT reagent. The TMT126 reagent in each 10-plex was reserved for a bridge channel, which consists of an equal amount of each sample pooled together, and the remaining TMT reagents were used to label individual sample digests. The bridge channel served to control for experimental variation between individual 10-plex experiments. TMT labeling was conducted for one hour at RT, quenched with 9 μl of 5% hydroxylamine for 15 min at RT, then acidified with 50 μl of 1% TFA and pooled. The multiplexed samples were desalted as above to remove unreacted TMT reagents, then lyophilized.

**Basic reverse-phase liquid chromatography fractionation (bRPLC).** Multiplexed samples were fractionated by bRPLC with fraction combining as previously described[90]. Samples were resuspended in 5% formic acid in 5% acetonitrile and separated on a 4.6 mm × 250 mm C18 column using an Ultimate 3000 HPLC into 96 fractions. The resultant fractions were then combined into 24 fractions and lyophilized prior to LC-MS3 analysis.

**LC-MS3 analysis.** Samples were resuspended in 5% acetonitrile/5% formic acid and separated on an Easy-nanoLC 1000 in-line with an Orbitrap Fusion Tribrid mass spectrometer. Samples were loaded onto a glass capillary column (length: 30 cm, I.D. 100 μm, O.D. 350 μm) pulled and packed in-house with 0.5 cm of 5 μm C4 resin followed by 0.5 cm of 3 μm C18 resin, with the remainder of the column packed with 1.8 μm of C18 resin. Once the sample was loaded, peptides were eluted using a gradient ranging from 11–30% acetonitrile in 0.125% formic acid over 180 min at a flow rate of 300 nl/min. The column was heated to 60 °C and electrospray ionization was achieved by applying of 2000 V of electricity through a T-junction at the inlet of the column.

All data were centroided and collected in data-dependent mode. An MS1 survey scan was performed over a mass to charge ($m/z$) range of 500–1200 at a resolution of 60,000 in the Orbitrap. Automatic gain control (AGC) was set to 200,000 with a maximum ion inject time of 100 ms and a lower threshold for ion intensity of 50,000. Ions selected for MS2 analysis were isolated with a width of 0.5 m/z in the quadrupole and fragmented using collision-induced dissociation (CID) with a normalized collision energy of 30%. Ion fragments were detected in the linear ion trap with the rapid scan rate setting with an AGC of 10,000 and a maximum inject time of 35 ms. MS3 analysis was conducted using the synchronous precursor selection (SPS) to simultaneously isolate 10 ions (regular proteomics) or 3 ions (phosphoproteomics) to maximize TMT sensitivity[95]. TMT reporter ions were fragmented off the peptides with higher energy collision induced dissociation (normalized energy of 50%) and MS3 fragment ions were analyzed in the Orbitrap at a resolution of 60,000. The AGC was set to 50,000 with a maximum ion injection time of 150 ms. MS2 ions 40 m/z below and 15 m/z above the MS1 precursor ion were excluded from MS3 selection.

**Data processing and analysis.** Resultant mass spectrometry data files were analyzed using Proteome Discoverer 2.1. MS2 spectra were queried against the Uniprot human protein database (downloaded: 05/2017) using the Sequest search algorithm[96]. A decoy search was also conducted with sequences in reverse to estimate FDR[97–99]. A mass tolerance of 50 ppm was used for MS1 spectra and a tolerance of 0.6 Da was used for MS2 spectra. TMT 10-plex reagents on lysine and peptide n-termini and carbamidomethylation of cysteines were included as static modifications. Oxidation of methionine and, for the phospho-proteomics experiments, phosphorylation of serine, threonine, and tyrosine residues, were also included in the search parameters as variable modifications. The target-decoy strategy was used to filter results to a 1% FDR at the peptide and protein levels[97–99]. Reporter ion intensities extracted from MS3 spectra were used for quantitative analysis. For regular proteomics, protein-level abundance values were calculated by summing signal-to-noise values for all peptides per protein meeting the specified filters. Data were normalized as previously described[100]. Phosphopeptide abundance was normalized similarly, except quantitation was summed to the unique phospho-peptide level then normalized to the total protein level. Phosphosite localization was performed using the PhosphoRS node within Proteome Discoverer. The PTMphinder R package was used to localize phosphorylated residues in the context of full-length proteins[101]. The significant peptides were determined via "pi score" which combines $t$ test $p$ values and fold-changes into a single metric[102] (PAPER XIAO 2014, Bioinformatics). We used a pi score that corresponds to an alpha of <0.05 as a cutoff for significant proteins. The $t$ test used to calculate the pi score was two-sided and not corrected for multiple hypotheses. Prior to direct statistical comparisons, K-means clustering was used to group all quantified phosphopeptides with similar expression profiles. Gene ontology analysis was used to identify enriched pathways in clustered phosphopeptides through the DAVID server (https://david.ncifcrf.gov/)[103,104]. K-means clustering was used to group all quantified phosphopeptides with similar expression profiles, prior to direct statistical comparisons. STRING-db (v11.0) was utilized to generate functional protein association networks of proteins of interest[105]. Connections were limited to high confidence (0.7) with a maximum of 20 connections for the second shell. First shell of interactions was restricted to the query only.

## Spine density analysis

**Brain slice preparation.** Mice were deeply anesthetized with keta-mine, and perfused with 0.9% w/v Sodium Chloride, then perfused with 4% paraformaldehyde (PFA) in phosphate buffer (PB). The brain tissues were removed and post-fixed in 4% PFA in PB for 30 min. Using a vibratome, 100 μm coronal sections were sliced and stored in 1× dPBS. Alexa Fluor® 594 Hydrazide (Thermo Fisher Scientific A10438) was injected into CA1 pyramidal neurons to follow neuronal projections. Injected slices were post-fixed for 15 min on 4% PFA in PB prior to mounting with Aqua-Poly/Mount (Polysciences Inc. 18606-20).

**Confocal microscopy and dendritic spine analysis.** Immuno-fluorescent images of hippocampal neurons were acquired with a Leica DMI6000 inverted microscope equipped with a Yokogawa Nipkow Spinning disk confocal head, an Orca ER High-Resolution black and white cooled CCD camera (6.45 μm/pixel at 1×) (Hamamatsu), Plan Apochromat 63×/1.4 numerical aperture objective, and Andor 100 mW 561 nm laser. Confocal z-stacks were acquired in all experiments and all imaging was acquired in the dynamic range of 8-bit acquisition (0–255 pixel intensity units, respectively) with Volocity v6.1.1 (PerkinElmer) imaging software. Imaged dendrites from one secondary dendrite per cell (after 1 branch) at a distance of 40–80 μm from the soma were straightened using ImageJ v1.41o. We estimated spine density as the number of manually counted spines (length 2 experimental conditions). Statistical significance was determined using unpaired Student's $t$-tests.

## Western blot analysis

Fresh frozen post-mortem human brains (frontal cortex) from long-itudinally followed and characterized individuals were provided by the ADRC Neuropathology Core at UCSD. Mice were euthanized and brain samples were obtained and snap-frozen immediately after collection. All procedures involving animals were approved by The Scripps Research Institute's Institutional Animal Care and Usage Committee (IACUC) and the UCSD IACUC, and met the guidelines of the National Institute of Health detailed in the Guide for the Care and Use of Laboratory Animals[81].

Frozen brain tissues were lysed and homogenized in a Dounce tissue grinder with RIPA buffer (50 mM Tris, pH 7.4, 1% Triton X-100, 1% NaDOC, 0.1% SDS, 150 mM NaCl, 2 mM EDTA, 10 mM NaF, 1 mM DTT, 1 mM $Na_3VO_4$, 1 mM PMSF, 50 μg/mL leupeptin, 1 μM microcystin, and 2 mM benzamidine). Homogenates were sonicated and protein was quantified using a BCA protein assay kit (Thermo Fisher Scientific). Fifty micrograms of protein were separated by standard SDS/PAGE and transferred to PVDF membranes (BioRad). Membranes were blocked with 5% BSA or 5% milk for one hour at room temperature and analyzed by immunoblotting with specific antibodies. Detection of immunoreactive bands was performed via chemiluminescence on a FluorChemQ imaging system (Alpha Innotech). Analyses were performed using the AlphaView SA, version 3.4.0 software.

Antibodies: anti-PKCα antibody (610108, clone 3/PKCα, used at 1:1,000 dilution) was from BD Transduction Laboratories. β-Actin antibody was purchased from Sigma-Aldrich (A2228, clone AC-74, used at 1:20,000 dilution), total SAP97 was from Enzo Life Sciences (ADI-VAM-PS005, clone RPI 197.4, used at 1:1000 dilution), phospho-MARCKS (sc-12971-R) and total MARCKS (sc-6454) were obtained from Santa Cruz Biotechnology (both used at a 1:250 dilution). GAPDH (2118, clone C1410), vinculin (4650), phospho-(Ser) PKC substrate (2261 S), phospho-ERK1/2 (9101) and total ERK1/2 (9102) antibodies were purchased from Cell Signaling Technology, and were used at a 1:1000 dilution. The pSAP97 (T656) antibody was custom made by NeoMPS by immunizing rabbits with an Ac-CKERARLK-T(PO3H2)-VKFN-NH2 peptide that was conjugated to keyhole limpet hemocyanin and was affinity-purified using the phosphopeptide antigen, as described and characterized in O'Neill *et al.*[32]. It was used at a 1:1000 dilution.

## Statistical analysis

For the statistical analyses of behavioral results, analysis of variance (ANOVA) was used followed by post hoc Student's *t* tests as appropriate. For spine density statistical analysis and immunoblots statistical analysis unpaired Student's *t* tests were used. To assess statistical significance of dual-patch recordings, two-way ANOVA was used followed with Newman-Keuls multiple comparison post hoc test (to compare CT100-induced depression in WT versus M489V mice). Paired *t* tests were used to assess CT100-induced depression in dual-recordings performed simultaneously in WT and M489V mice. Data was graphed using GraphPad Prism 9.

## Reporting summary

Further information on research design is available in the Nature Portfolio Reporting Summary linked to this article.

## Data availability

The MS proteomics data generated for this manuscript, including annotated spectra, have been deposited onto the ProteomeXchange archive through MassIVE under the following identifiers: PKC-M489V 3 months old mice, C57BL/6 N background (ProteomeXchange: PXD029092), PKC-M489V 4.5 and 6 months old mice, APP transgenic background (ProteomeXchange: PXD029093). The processed MS proteomics are available within the Dataset 1 and Dataset 2 files. All other data needed to evaluate the conclusions in the paper are contained within the manuscript or the Supplementary Information file and the Source Data file. Source data are provided with this paper.

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

## Acknowledgements

We thank the members of the Newton laboratory and the laboratory of Dr. Jack E. Dixon for helpful discussions and Dr. Roberto Malinow for advice on the electrophysiological approaches. We also thank the Collaborative Center of Multiplexed Proteomics at UCSD for the

phosphoproteomics analysis, Dr. Donald Pizzo from the UC San Diego Human and Animal Tissue Technology Center for processing the histology samples, and Dr. Robert Rissman and Jeffrey Metcalf from the Alzheimer's Disease Research Center (ADRC) Neuropathology Core and Brain Bank at UCSD for providing human brain samples, which were collected with support from the NIH UCSD ADRC Center Grant (NIH AG062429 to RR). The cartoon in Fig. 3a was created with BioRender.com. This work was supported by Cure Alzheimer's Fund (A.C.N. and R.E.T.) and NIH R35 GM122523 (A.C.N.). J.M.W. was supported by the University of California, San Diego Graduate Training Programs in Cellular and Molecular Pharmacology (T32 GM007752) and Rheumatic Diseases Research (T32 AR064194).

## Author contributions

G.L., J.M.W., K.D., L.E.D, A.J.R., and C.C.-G. performed the experiments. J.M.W. performed the MS analysis mentored by D.J.G. K.D. performed the electrophysiology experiments. L.E.D. performed the spine density analysis mentored by G.N.P. A.J.R. and C.C.-G. performed the behavioral experiments. R.E.T. coordinated the generation of the mouse models. G.L. and A.C.N. conceived the project, designed the experiments, and wrote the manuscript. All authors edited the manuscript.

## Competing interests

The authors declare no competing interests.
