## [Peer Review File · Nature Communications]

Enhanced Activity of Alzheimer Disease-associated Variant of Protein Kinase C α Drives Cognitive Decline in a Mouse ModelREVIEWER COMMENTS

Reviewer #1 (Remarks to the Author):

This is a superb study in which the authors undertake thorough examination of a gain of variant in one isoform of PKCalpha (M489V) that is associated with Alzheimer's disease. The authors undertake a variety of analyses to characterise this knock-in mouse strain and compare it with wild type as well as heterozygous animals. They demonstrate that the PKCalpha M489V variant markedly enhances the phosphorylation of many proteins in the brain, including well characterised PKC substrates such as MARCKS. Validation studies using immunoblotting are performed. The bioinformatic analysis suggests that proteins involved in post-synaptic density synapse cell junction and other processes of relevance to understanding Alzheimer's disease, are impacted in the knock-in mice expressing the active PKC variant. The applicants also demonstrate that the PKC variant induces loss of spine density in hippocampal neurons, and perform trained maze analysis to establish that the PKCalpha M489V animals display significant impairment of cognition. The authors have also crossed the PKC M489V variant into the APP transgene mouse model of Alzheimer's disease that results in elevated amyloid beta proteins. The authors show that levels of APP are elevated as expected in the APP mice and that levels of PKC expression were not altered by the M489V mutation. Significantly, enhanced APP expression was associated with a significant increase in PKC activity as seen by enhanced phosphorylation of PKC substrates and this was further impacted by the M489V mutation. They also describe a set of proteins whose phosphorylation is substantially increased in aged mice harbouring the M489V mutation that are not enhanced in younger mice. The authors also demonstrate that the PKC alpha and M489V mice accelerate impaired cognition in the APP mouse model, and by 6-months of age the mice bearing APP and the M489V mutation lost the ability to discriminate between target quadrants, indicating significant loss of cognition. Furthermore, the authors demonstrate that the PKC alpha M489V variant does not impact amyloid beta levels in the brain of APP mouse models but significantly enhances synaptic depression. The authors provide some data to suggest that PKC levels are also increased in the brain of human patients with Alzheimer's disease.

The paper is extremely well written, and the results are clear-cut and the figures carefully presented. A lot of control studies are performed for each set of experiments and the data are robust and persuasive. I believe that this study provides a significant advance in our understanding of how increased expression or activity of PKC is linked to Alzheimer's disease. Groups working on Alzheimer's disease as well as kinase biology would be interested in this study. It also provides data that indicates that strategies to reduce PKC alpha activity could be considered as a possible treatment for Alzheimer's disease. The mouse models described in this study would be perfect models for further testing and validating such therapeutic strategies. I recommend that this paper be accepted. Below I list some minor points that the authors could consider in their revised version of the manuscript:

- 1. Can the authors deposit the full mass spectrometry and phosphoproteomic data analysis on the PRIDE database, which would enable others to better exploit and re-analyse their data?**
- 2. Is it possible for the authors to include a table of the top 25 phosphosites that are enhanced by the PKC M489V variant, showing the phosphorylation sites and encompassing residues-Do this both for the data in Fig 1 and 4? This could maybe go in a supplementary table.**
- 3. In Fig.4e the authors perform string analysis on the top phosphoproteomic hits. Is it worth performing similar analysis on the phosphoprotein hits characterised in Fig.1, so the results could be compared more easily?**
- 4. The data in fig.8 with human frontal cortex brain samples, are these collected from**

the same clinical centre and were the samples generated and isolated in the same manner? Presumably all consent issues for inclusion of this data have been obtained by the authors.

5. The authors claim that the M498V variant should enhance PKC alpha activity by around 30% which is consistent with the data presented in the study. Is it possible for the authors to immunoprecipitate the wild type and variant PKC from a mouse brain extract and undertake kinase assays to demonstrate that the variant does indeed enhance the PKC kinase activity?

6. I would suggest that the half page of materials and methods that are present in the supplementary method be moved to main methods section, that is extremely comprehensive and well prepared. I do not see the reason for including these methods in the supplementary section and not in the main section.

Reviewer #2 (Remarks to the Author):

Lordén et al. report a nice and broad multi-method analysis of a mouse modeling the AD-associated PKC variant M489V. The Reviewer considered the study to have several strengths including an important topic of broad interest, diverse techniques support the proposed conclusions, and a generally well written and understandable narrative. However, several substantial critiques were noted, and the current version is further weakened by several unnecessary data inconsistencies and overstatements. Most of the data is of high quality but too many of plots lack robust statistical analysis, plus the order of the figures, and main versus supp figs distribution needs attention. The authors show that the PKC variant M489V remodels the whole brain phosphoproteome. They quantify overall protein fold change too, I am not sure if and how these measures may have been integrated though. They go on to suggest that this alteration causes synaptic dysfunction and drives cognitive decline - this is substantial claim especially as stated in the title. The mass spectrometry (MS) experiments were well-performed but there are limitations to the current datasets as presented that could be considered. There is also a big gap between MS results and behavioral data. Moreover, the Fig.6 shows negative results and Fig. 8 is quite disconnected from all other figures.

Overall I sense there is something interesting to this body of work - but a more consistent and coherent (for lack of a better word) set of figure panels needs to be presented. If the authors can address these concerns with new experiments and data reorganization, I could be motivated to strongly support this manuscript for pages in Nat Comm.

Major points:

1. Regarding Fig. 1 and Fig. 4, Is there any reason to use whole brains for TMT-based phosphoproteomics analyses? Hippocampus and cortex are the major brain regions for AD pathology, other regions such as the cerebellum are far and away less much less affected by AD. Mixture of all brain regions will dilute the true signals. In other words, if unaffected brain regions dominates the phosphoproteomic analysis the phenomenon may not important for AD pathology. Can the analysis possible be repeated in cortical extracts? MARKs may even emerge as an even more significant candidate. The Reviewer find this to point to be important.

2. Fig. 1D showing there is a dosage decrease in WT/WT, WT/M489V and M498V/M489V brains. How do the authors imagine the enhancement of PKC activity lead to decrease of protein phosphorylation? This result at least suggests that indirect or compensatory effects are predominant.

3. Since Fig. 1E suggests there is a dosage increase in WT/WT, WT/M489V and

M498V/M489V brains, Fig. 1C and all the panels in Fig.2 should include data from WT/M489V brain. The details are not critical but this point illustrates the disconnect between the figure panels.

4. Fig. 3 should include behavioral data of WT/M489V mice. Moreover, could PKC inhibitors rescue this behavioral deficit? I realize this is a major ask, but this point should be considered.

5. For Fig.4C, is there any difference between M498V and M489Vapp samples? Why only report WT vs WTapp? The Reviewer is include to think they may not have supported the hypothesis. Please consider removing or de-emphasizing contradictory or extraneous panels (for example old mice used in the behavior experiments).

6. For Fig. 4D, A suggestion - could you show some volcano plots to compare M498V and M489Vapp?

7. Fig.5 also could benefit rescue experiments using PKC inhibitor to support the hypothesis that cognitive deficits in AD mice are caused by enhanced PKCM498V kinase activity. Not a required experiment but could provide key support to the conclusions.

8. Is the phosphorylation level higher in 12-month-old mouse brain than other younger mice ?

9. Fig.6 shows negative results and should be moved to the supp and introduced earlier in the narrative.

10. There is a big gap between the electrophysiological data and MS data and behavioral data. Why not examine synaptic plasticity (e.g., LTP) should be examined in WTapp and M489Vapp mice? The Abeta experiment just hangs, if necessary, it could be removed. These results speak to only synaptic transmission properties, no?

11. Fig. 8, I didn't see any data about phosphorylated SAP97 in previous figures. This whole figure is not well-connected with all other seven figures.

12. The description of the APP transgenic mice misses the mark. The nomenclature needs attention, I suggest tg-AD and non tg-AD for example. This is an issue throughout the manuscript. The senior authors may be able to easily help with this point.

13. Love the TMT / phosphor experiments, but I am not clear on how / if protein abundances were incorporated. Also, there is a "compensational" issue here, since the tmt labeling happened after enrichment and the multiplexing was performed based on equal input material or if not then dissimilar labeling efficiency can be an issue. Have the authors examined their tmt labeling efficiency?

14. WB in Figure 2C looks like it may be limited by an unequal transfer. The axis labels need major attention and the units are vague and normalization is poorly defined. Should turn up B/C in fig 2a. Stats needed in all plots.

Minor points:

- 1. Have you compared WT/M489V to M498V/M489V or WT/WT to WT/M489V using volcano plot?**
- 2. The authors should use phosphatase (e.g., lamda PP) treatment to conform the bands in Fig. 2C are phosphorylated proteins.**
- 3. The representative images look very similar.**
- 3. Did you perform the open field and water maze assays?**
- 4. Have you tested phosphorylation of tau in WTapp and M489Vapp mice?**
- 5. Some figures (e.g., Fig. 3B-E) are bar plots. Individual data points should be displayed as in Fig. 4B.**
- 6. Please check and consider lines 48 – abeta 42 peptides, line 55 plaques and**

development are not well aligned, lines 70-75 maybe condense, no need to dwell on cancer, line 116 FDR at what level PSM, peptide, protein?, line 122 10% of phosphopeptides are sig? that's major, line 137- AD mutation? Too strong, line 153- can predict?,

Reviewer #3 (Remarks to the Author):

In this ms by Lordén et al, the authors have addressed the question of an AD-associated mutation located in PKCa found in a few families affected by LOAD. The authors produce mice with a PKCaM489V mutation. The authors found that the PKCa mutation changed the p-proteome including some substrates of PKCa, leading to a slight decreased spine density, alterations in memory in mice with the PKCa mutation, which was further aggravated in APP2576 mice, without affecting abeta levels. Finally, data from AD brains revealed changes in the PKCa levels. These are potentially interesting and important for the AD field.

The ms is clearly written, the data as presented fairly robust, and the conclusions drawn appropriate. However, additional analysis is required to strengthen the hypothesis put forward. These include careful assessment of memory and cognition by several tests in order to support the authors claim that PKCa drives cognitive decline in AD as postulated in the title. Below are points listed that the authors need to consider.

1. Please provide evidence showing the successful insertion of the PKCa mutation in the edited genome e.g. sequencing data.
2. p-values in Fig 1c missing.
3. Mutation-induced loss of spines is limited though significant, please substantiate this finding by analyzing synaptic markers such as PSD95 and synaptophysin with wb and IHC assessments. EM could also show a decrease in synapses.
4. WB in Fig 2C upper panel where substrates for PKCa have been probed for is not convincing, the proteins are not evenly blotted to the membrane, please repeat and provide a representative blot.
5. The behavioral analysis needs to be further elaborated. Additional basic memory tests addressing hippocampal-dependent memory should be used e.g. Y-maze, morris water maze, fear conditioning.
6. Fig legend Fig 4 F does not match the images. Please quantify wb data. Has statistical analysis been performed of the pERK p-peptides presented in the graph? What was the outcome?
7. Abeta analysis, please also provide IHC evidence of no change in abeta plaque pathology. These data and abeta ELISA data can be presented in Supp mat.
8. For e-phys measurements, to avoid the side effects of viral expression of CT100, please measure the e-phys properties of acute slices from the mice incubated with abeta.
9. Please describe in all the fig legends what statistical analysis have been used.

Response Letter: **Enhanced Activity of Alzheimer Disease-associated Variant of Protein Kinase α Drives Cognitive Decline**

All revisions are **highlighted in yellow** in the revised manuscript.

REVIEWER COMMENTS

Reviewer #1 (Remarks to the Author):

This is a superb study in which the authors undertake thorough examination of a gain of variant in one isoform of PKC α (M489V) that is associated with Alzheimer's disease. The authors undertake a variety of analyses to characterise this knock-in mouse strain and compare it with wild type as well as heterozygous animals. They demonstrate that the PKC α M489V variant markedly enhances the phosphorylation of many proteins in the brain, including well characterised PKC substrates such as MARCKS. Validation studies using immunoblotting are performed. The bioinformatic analysis suggests that proteins involved in post-synaptic density synapse cell junction and other processes of relevance to understanding Alzheimer's disease, are impacted in the knock-in mice expressing the active PKC variant. The applicants also demonstrate that the PKC variant induces loss of spine density in hippocampal neurons, and perform trained maze analysis to establish that the PKC α M489V animals display significant impairment of cognition. The authors have also crossed the PKC M489V variant into the APP transgene mouse model of Alzheimer's disease that results in elevated amyloid beta proteins. The authors show that levels of APP are elevated as expected in the APP mice and that levels of PKC expression were not altered by the M489V mutation. Significantly, enhanced APP expression was associated with a significant increase in PKC activity as seen by enhanced phosphorylation of PKC substrates and this was further impacted by the M489V mutation. They also describe a set of proteins whose phosphorylation is substantially increased in aged mice harbouring the M489V mutation that are not enhanced in younger mice. The authors also demonstrate that the PKC α and M489V mice accelerate impaired cognition in the APP mouse model, and by 6-months of age the mice bearing APP and the M489V mutation lost the ability to discriminate between target quadrants, indicating significant loss of cognition. Furthermore, the authors demonstrate that the PKC α M489V variant does not impact amyloid beta levels in the brain of APP mouse models but significantly enhances synaptic depression. The authors provide some data to suggest that PKC levels are also increased in the brain of human patients with Alzheimer's disease.

The paper is extremely well written, and the results are clear-cut and the figures carefully presented. A lot of control studies are performed for each set of experiments and the data are robust and persuasive. I believe that this study provides a significant advance in our understanding of how increased expression or activity of PKC is linked to Alzheimer's disease. Groups working on Alzheimer's disease as well as kinase biology would be interested in this study. It also provides data that indicates that strategies to reduce PKC α activity could be considered as a possible treatment for Alzheimer's disease. The mouse models described in this study would be perfect models for further testing and validating such therapeutic strategies. I recommend that this paper be accepted. Below I list some minor points that the authors could consider in their revised version of the manuscript:

1. Can the authors deposit the full mass spectrometry and phosphoproteomic data analysis on the PRIDE database, which would enable others to better exploit and re-analyse their data?

The full mass spectrometry and phosphoproteomic data sets have been uploaded on the MassIVE data base. We have added this information to the Data availability section, p. 24 (line 623):

C57 background experiment (PXD029092/MSV000088224):

<https://massive.ucsd.edu/ProteoSAFe/dataset.jsp?task=f83e635fd21544718978b86056d6daca>

APP transgenic mice experiment (PXD029093/MSV000088225):

<https://massive.ucsd.edu/ProteoSAFe/dataset.jsp?task=f4bda1f37ce445128e83b8133f7f3da0>

The reviewers can follow the instructions after clicking the link and use the password "G0nza13Z" to access the data.

2. Is it possible for the authors to include a table of the top 25 phosphosites that are enhanced by the PKC M489V variant, showing the phosphorylation sites and encompassing residues-Do this both for the data in Fig 1 and 4? This could maybe go in a supplementary table.

We have added **new Supplementary Table 1** and **new Supplementary Tables 2 and 3** showing the top 25 phosphosites enhanced in the brain samples from the PKC α M489V compared with WT either on the WT background (Supp Table 1) or the on tg-AD and tg-AD background (Supp Tables 2 and 3). See comments in p.6 (line 126) and p.10 (line 252).

3. In Fig.4e the authors perform string analysis on the top phosphoproteomic hits. Is it worth performing similar analysis on the phosphoprotein hits characterised in Fig. 1, so the results could be compared more easily?

We performed the same string analysis of the data in Figure 1 (so comparing M489V to WT PKC on a WT background) as we did for the data in Figure 4e (now 5e) comparing the M489V to WT PKC on the AD background. Interestingly, whereas the M489V mutation impacted mTOR, MAPK and Neuron projections for the mice on the AD background, the mTOR and MAPK were considerably less impacted on the WT background, but Neuron projections were highly impacted (see graph below). This suggests the M489V synergizes with the AD defect to impact mTOR and MAPK, but alone, its biggest impact is on neuron projections (consistent with data such as the spine density in Figure 2A). This is also consistent with the analysis in Figure 1D showing the gene dosage effects on synapses and cytoskeleton caused by the M489V. We have added this text (p. 11, line 71):

note that neuron projection processes were also significantly perturbed by the M489V variant on a WT background (e.g. **Figure 1D,E**).

4. The data in fig.8 with human frontal cortex brain samples, are these collected from the same clinical centre and were the samples generated and isolated in the same manner? Presumably all consent issues for inclusion of this data have been obtained by the authors.

All the samples were collected from the ADRC Neuropathology Core at UCSD were processed in the same way. We have consent to include these data in the manuscript.

5. The authors claim that the M498V variant should enhance PKC alpha activity by around 30% which is consistent with the data presented in the study. Is it possible for the authors to immunoprecipitate the wild type and variant PKC from a mouse brain extract and undertake kinase assays to demonstrate that the variant does indeed enhance the PKC kinase activity?

Previous characterization of PKC α and PKC α M489V expressed in insect cells and purified to homogeneity revealed that the activity of the variant is approximately 30% higher than that of WT enzyme¹; this result was also observed in cells overexpressing WT vs M489V using our genetically encoded reporter CKAR². We have now validated that this is also the case for the PKC α in the mouse model: *in vitro* kinase assays of PKC α immunoprecipitated from 5 WT brains or 5 M489V brains revealed higher Ca²⁺/lipid-stimulated activity in enzyme immunoprecipitated from the M489V brains compared with WT. These data are now in **new Supplementary Figure 1G** and discussed in the Results (p. 6, lines 129-132).

6. I would suggest that the half page of materials and methods that are present in the supplementary method be moved to main methods section, that is extremely comprehensive and well prepared. I do not see the reason for including these methods in the supplementary section and not in the main section.

Out of concern for the word limit, we have not moved the materials and methods from the Supplement to the main Methods (note to editor: if this is not a problem, we would be pleased to move to main text).

Reviewer #2 (Remarks to the Author):

Lordén et al. report a nice and broad multi-method analysis of a mouse modeling the AD-associated PKC variant M489V. The Reviewer considered the study to have several strengths including an important topic of broad interest, diverse techniques support the proposed conclusions, and a generally well written and understandable narrative. However, several substantial critiques were noted, and the current version is further weekend by several unnecessary data inconsistencies and overstatements. Most of the data is of high quality but too many of plots lack robust statistical analysis, plus the order of the figures, and main versus supp figs distribution needs attention. The authors show that the PKC variant M489V remodels the whole brain phosphoproteome. They quantify overall protein fold change too, I am not sure if and how these measures may have been integrated though. They go on to suggest that this alteration causes synaptic dysfunction and drives cognitive decline - this is substantial claim especially as stated in the title. The mass spectrometry (MS) experiments were well-performed but there are limitations to the current datasets as presented that could be considered. There is also a big gap between

MS results and behavioral data. Moreover, the Fig.6 shows negative results and Fig. 8 is quite disconnected from all other figures.

Overall I sense there is something interesting to this body of work - but a more consistent and coherent (for lack of a better word) set of figure panels needs to be presented. If the authors can address these concerns with new experiments and data reorganization, I could be motivated to strongly support this manuscript for pages in Nat Comm.

Major points:

Major points:

1. Regarding Fig. 1 and Fig. 4, Is there any reason to use whole brains for TMT-based phosphoproteomics analyses? Hippocampus and cortex are the major brain regions for AD pathology, other regions such as the cerebellum are far and away much less affected by AD. Mixture of all brain regions will dilute the true signals. In other words, if unaffected brain regions dominates the phosphoproteomic analysis the phenomenon may not be important for AD pathology. Can the analysis be repeated in cortical extracts? MARKs may even emerge as an even more significant candidate. The Reviewer find this to be important.

The reviewer raises a good point. The reason we used whole brain was that we were concerned that we would not obtain sufficient protein from hippocampus to get high quality phosphoproteomics data. For this reason, we instead interrogated substrates such as MARCKS and ERK by western blot analysis of extracts from hippocampus (Figure 2).

2. Fig. 1D showing there is a dosage decrease in WT/WT, WT/M489V and M498V/M489V brains. How do the authors imagine the enhancement of PKC activity lead to decrease of protein phosphorylation? This result at least suggests that indirect or compensatory effects are predominant.

Mounting evidence suggest that a primary target of PKC is to control the phosphatase output of cell. Thus, enhanced activity of PKC would enhance the phosphatase output, leading to a reduction in the phosphorylation state of certain substrates. It is also possible that PKC is inhibiting the activity of kinases directed at these substrates. We note on line 149-152 (p.6-7):

“These data are consistent with enhanced PKC α function increasing the phosphorylation of direct substrates such as MARCKS to modulate cytoskeletal function, and indirectly decreasing the phosphorylation of substrates that are key regulators of the synapse, either by enhancing phosphatase activity or inhibiting kinases directed at these substrates.”

3. Since Fig. 1E suggests there is a dosage increase in WT/WT, WT/M489V and M498V/M489V brains, Fig. 1C and all the panels in Fig.2 should include data from WT/M489V brain. The details are not critical but this point illustrates the disconnect between the figure panels.

The gene dosage plots involve hundreds of phosphopeptide allowing trends to be quite obvious. Given we only have 3 data points for a specific phosphopeptide, the intermediate change of the HETS is within the size of the error bars so we prefer not to include. We include the data for the reviewer for the pMARCKS (S163) as an example:

4. Fig. 3 should include behavioral data of WT/M489V mice. Moreover, could PKC inhibitors rescue this behavioral deficit? I realize this is a major ask, but this point should be considered.

The behavioural data were a massive undertaking leading us to focus on the WT vs HOM, on the two backgrounds, at the three ages. With regards to the rescue by PKC inhibitors, we fully agree. Although treating the animals with

PKC inhibitors is beyond the scope of this study, we are in the process of developing antisense oligos for PKC α for this purpose.

5. For Fig. 4C, is there any difference between M498V and M489Vapp samples? Why only report WT vs WTapp? The Reviewer is include to think they may not have supported the hypothesis. Please consider removing or de-emphasizing contradictory or extraneous panels (for example old mice used in the behavior experiments).

(Note now Figure 5C). The question driving the analysis of the WT vs tg-AD was whether PKC activity is higher in the AD mouse model. This is the same question we ask in the analysis of human brain from control vs AD patients (Figure 7C). We show that enhanced PKC activity is associated with both mouse models of AD (Figure 5C) and human AD (Figure 7C). This is important because it reveals that enhanced PKC activity may generally be associated with the development of AD and is not just restricted to the patients that have the M489V mutation. Deregulation of any of the abundance of mechanisms that control the steady state levels of PKC to increase steady-state levels of PKC, as we showed in our previous study ¹, may occur in AD, identifying PKC as a potential therapeutic target in AD generally (and not just for the rare patients with the M489V mutation). We note in Discussion (line 359-362, p. 14):

“Furthermore, analysis of post-mortem human brain revealed that, specific mutations aside, patients with AD generally have higher steady-state levels of PKC α . This suggests that not only are mutations in PKC α biomarkers for the disease, but the intrinsic set point of PKC α levels may also predict susceptibility to AD.”

6. For Fig. 4D, A suggestion - could you show some volcano plots to compare M498V and M489Vapp?

We present the volcano plot of M489V and M489Vapp below. As the manuscript is extremely data-heavy, we prefer not to include as it has not unveiled any key information not already presented.

7. Fig.5 also could benefit rescue experiments using PKC inhibitor to support the hypothesis that cognitive deficits in AD mice are caused by enhanced PKC α kinase activity. Not a required experiment but could provide key support to the conclusions.

As noted above, our long term plan is to do exactly as the reviewer suggests – probe whether PKC inhibitors can reverse the behavioral deficit on the M489V mice, either on the WT or app background.

8. Is the phosphorylation level higher in 12-month-old mouse brain than other younger mice?

We did check by western blot the phosphorylation of different substrates in mice as a function of age. Curiously, the phosphorylation on Tyrosine increased with age, and the phosphorylation on Serine seemed to increase at 6 months and decrease at 12 months (see figure below). As the contribution of phosphatase regulation in ageing is understudied, it is difficult to draw meaningful conclusions from these blots. This information is tangential to our story (which focusses on the effect of enhanced PKC activity, rather than how age affects phosphorylation) so we prefer not to include it.

9. Fig.6 shows negative results and should be moved to the supp and introduced earlier in the narrative.

Fig 6 has been moved to the supplement (now **new Supplementary Figure 3A-C**) and is introduced earlier in the narrative (p.9, lines 221-229), as suggested. Specifically, we have reorganized the manuscript to first present the electrophysiology data of the brain slices from the WT vs M489V mice (now section 4. starting on line 195), and then introduce the abeta data right at the beginning of the use of the tg-AD mouse model (now section 5. starting line 213).

10. *There is a big gap between the electrophysiological data and MS data and behavioral data. Why not examine synaptic plasticity (e.g., LTP) should be examined in WTapp and M489Vapp mice? The Abeta experiment just hangs, if necessary, it could be removed. These results speak to only synaptic transmission properties, no?*

Indeed, the experiment we performed addressed the question: does Abeta have a greater effect on excitatory synaptic transmission in the M489V mice than on WT mice? And we show the key result that excitatory transmission in the M489V mice is extremely sensitive to Abeta – with a striking 55% reduction in the M489V mice compared to the WT. Since the M489VAPP mice show lower memory scores, this would be consistent with lower excitatory synaptic transmission in these animals (particularly in the hippocampus, which is likely required in the memory experiments done). Thus, the electrophysiological data are supportive of the behavioral data.

Regarding examining LTP of the mice on the app background, we did not focus on these given that plasticity is a non-linear process, and we observe a quite striking effect on both the electrophysiology (and the behavior) on the WT background. Meaning, the key result is that excitatory transmission in the M489V mice is extremely sensitive to beta amyloid in the absence of other perturbations. This striking 55% reduction of excitatory transmission suggests that the amount of hippocampal plasticity *in vivo* could be reduced enormously. Importantly, plasticity (like NMDAR activation) is a non-linear process. In the M489V animals with beta amyloid (*M489Vapp*), more than 2x the number of inputs, compared to wt animals, would need to be co-active to produce LTP. But we may not detect this using artificial depolarization during whole-cell LTP induction in slices from app mice because of this non-linearity. Specifically, early studies established that one of the fundamental properties of LTP is it displays a threshold (also called 'cooperativity') below which there is zero plasticity (e.g. see references ^{3, 4, 5}). Thus, by using artificial depolarization during whole-cell LTP induction in slices, there may not be differences in LTP in M489V animals on the APP background even though the amount of LTP occurring during learning protocols in these animals would be very compromised. As noted above, our key finding is this extreme sensitivity to beta amyloid, an effect easily unmasked in the WT background. Consistent with this, the behavioral data showed that at 12 months, the cognitive impairment of the WT app mice was so severe, there was no further decline in the M489V mice. We needed to examine behavior at younger ages to unmask a difference on the APP background (or at any age on WT background).

11. *Fig. 8, I didn't see any data about phosphorylated SAP97 in previous figures. This whole figure is not well-connected with all other seven figures.*

We have better clarified in the text that phosphorylation of SAP97 is yet another marker for PKC activation (p.13, starting line 321):

“Additionally, as another marker for PKC activation, we assessed the phosphorylation state of T656 on SAP97, a previously identified PKC phosphorylation site⁶. Phosphorylation of T656 was increased by approximately four-fold in the brains from AD patients compared to control patients, consistent with higher PKC activity (**Figure 7B**).”

12. *The description of the APP transgenic mice misses the mark. The nomenclature needs attention, I suggest tg-AD and non tg-AD for example. This is an issue throughout the manuscript. The senior authors may be able to easily help with this point.*

The nomenclature has been updated throughout the manuscript, as suggested by the reviewer, to non tg-AD and tg-AD mice.

13. Love the TMT / phosphor experiments, but I am not clear on how / if protein abundances were incorporated. Also, there is a “compensational” issue here, since the tmt labeling happened after enrichment and the multiplexing was performed based on equal input material or if not then dissimilar labeling efficiency can be an issue. Have the authors examined their tmt labeling efficiency?

As stated in the methods, prior to phospho-peptide enrichment, 50 ug of digested peptide was aliquoted for standard proteomic analysis and processed in parallel. After searching the proteomic data, phospho-peptide abundance was normalized to total protein levels.

We periodically check our labeling efficiency and routinely find it to be over 95%. We didn't examine the efficiency for these particular samples as it is not something we typically do for every sample. To ensure that TMT labeling is robust, we confirm the pH of each sample is ~8.5 prior to labeling (<https://pubs.acs.org/doi/10.1021/acsomega.1c00776>). Finally, the normalization we perform should account for any minor differences in labeling efficiency of individual labels.

14. WB in Figure 2C looks like it may be limited by an unequal transfer. The axis labels need major attention and the units are vague and normalization is poorly defined. Should turn up B/C in fig 2a. Stats needed in all plots.

We have replaced the blot with a better one in Figure 2C. The units 'relative phosphorylation' are phospho-signal of the relevant substrate divided by total signal of the substrate (e.g. pMARCKS/total MARCKS). We have clarified (Figure 2 legend, lines 984-990):

“Relative phosphorylation represents the densitometric analyses of the western blot phosphorylation signal and the total antibody signal of the indicated substrates. pERK1/2 (T202/Y187 for ERK1 and T185/Y187 for ERK2) and pMARCKS (S159/S163) signal was normalized to total ERK1/2 and total MARCKS signal respectively, and phospho-Ser PKC substrates signal was normalized to its GAPDH loading control. Data were normalized to the WT1 values, and normalized data from the depicted western blots were plotted as average normalized intensity \pm SEM.”

Minor points:

1. Have you compared WT/M489V to M498V/M489V or WT/WT to WT/M489V using volcano plot?

We have now added a volcano plot comparison of WT/M489V to M489V/M498V as well as WT/WT to WT/M489V in (new) **Supplementary Figure 1F** (WT vs HET and HET vs HOM).

2. The authors should use phosphatase (e.g., lambda PP) treatment to conform the bands in Fig. 2C are phosphorylated proteins.

We have used the phospho-Ser antibody in numerous previous studies and validated that it reads out PKC substrate phosphorylation: the signal is enhanced by when cells are treated with phorbol esters to activate PKC and suppressed with cells are treated with PKC inhibitors (e.g. see references ^{7,8}).

3. The representative images look very similar.

The quantification reveals a $9.81 \pm 0.02\%$ difference in spine density, from 1.25 ± 0.02 to 1.15 ± 0.02 spines/ μm (so for the 20 μm segment shown, an average of 25 vs 23 spines, which is likely to small to be detected by eye. This is why we quantified results from 125 dendritic segments (over 1500 spines).

3. Did you perform the open field and water maze assays?

We did not perform the water maze assay or open field test. However, we did perform the Y maze test and we did not observe significant differences caused by the PKC α variant at any of the ages tested. We also performed the nest building test, where we observed a robust nest-building deficit in the M489V mice at the age of 3 months. As this result was not observed in older mice, we did not include it. The Barnes Maze gave the most reproducible and statistically significant changes.

4. Have you tested phosphorylation of tau in WTapp and M489Vapp mice?

Yes, we probed for tau phosphorylation in these mice and did not see remarkable differences (See blot below).

5. Some figures (e.g., Fig. 3B-E) are bar plots. Individual data points should be displayed as in Fig. 4B.

We did not include these in the manuscript because the large number of data points obscures the bars and error bars. We include them below for the reviewer:

Figure 2A:

Figures 3A, 3B, 3C:

6. Please check and consider lines 48 – abeta 42 peptides, line 55 plaques and development are not well aligned, lines 70-75 maybe condense, no need to dwell on cancer, line 116 FDR at what level PSM, peptide, protein?, line 122 10% of phosphopeptides are sig? that's major, line 137- AD mutation? Too strong, line 153- can predict?,

We have made edits as suggested:

lines 48 – abeta 42 peptides – changed

line 55 plaques and development are not well aligned – rephrased to ‘the association between misprocessing and deposition of A β plaques and AD development’

lines 70-75 maybe condense, no need to dwell on cancer – prefer to leave these lines intact as they give the background on why cancer mutations are loss of function and very unusual to have a gain of function mutation

line 116 FDR at what level PSM, peptide, protein?, - we filtered for FDR < 1% at both the peptide and protein level. It has been added in the main text (line 117, p.5 and line 238, p.10).

line 122 10% of phosphopeptides are sig? that's major, - not sure what the reviewer is referring to, but, yes, these are significant changes.

line 137- AD mutation? Too strong – changed to AD 'variant'

line 153- can predict? – changed to 'is an indicator of'

Reviewer #3 (Remarks to the Author):

In this ms by Lordén et al, the authors have addressed the question of an AD-associated mutation located in PKCa found in a few families affected by LOAD. The authors produce mice with a PKCaM489V mutation. The authors found that the PKCa mutation changed the p-proteome including some substrates of PKCa, leading to a slight decreased spine density, alterations in memory in mice with the PKCa mutation, which was further aggravated in APP2576 mice, without affecting abeta levels. Finally, data from AD brains revealed changes in the PKCa levels. These are potentially interesting and important for the AD field.

The ms is clearly written, the data as presented fairly robust, and the conclusions drawn appropriate. However, additional analysis is required to strengthen the hypothesis put forward. These include careful assessment of memory and cognition by several tests in order to support the authors claim that PKCa drives cognitive decline in AD as postulated in the title. Below are points listed that the authors need to consider.

1. Please provide evidence showing the successful insertion of the PKCa mutation in the edited genome e.g. sequencing data.

We now provide sequencing data in the **new Supplemental Figure 1A** showing successful insertion of the PKCa mutation, and describe the method in Supplementary Materials and Methods on p.3 (lines 54-71).

2. p-values in Fig 1c missing.

p values have been added.

3. Mutation-induced loss of spines is limited though significant, please substantiate this finding by analyzing synaptic markers such as PSD95 and synaptophysin with wb and IHC assessments. EM could also show a decrease in synapses.

We did a western blot analysis of PSD95 in whole hippocampus and the synaptosome fraction in hippocampus. However, we were unable to see differences between WT and M489V. This is not surprising given the error of the assay is approximately 10% (see bar graph), and the difference we would be detecting is about 10%. Given pandemic restrictions and personnel issues, we were unable to do IHC or EM in a timely manner for the revision.

4. WB in Fig 2C upper panel where substrates for PKC α have been probed for is not convincing, the proteins are not evenly blotted to the membrane, please repeat and provide a representative blot.

We have replaced the blot in Figure 2C with a new one.

5. The behavioral analysis needs to be further elaborated. Additional basic memory tests addressing hippocampal-dependent memory should be used e.g. Y-maze, morris water maze, fear conditioning.

We did not perform the morris water maze assay or the fear conditioning test. As noted for Reviewer 2, point 3, we did perform the Y maze test and we did not observe significant differences caused by the PKC α variant at any of the ages tested. We also performed the nest building test, where we observed a robust nest-building deficit in the M489V mice at the age of 3 months. As this result was not observed in older mice, we did not include it. The Barnes Maze gave the most reproducible and statistically significant changes.

6. Fig legend Fig 4 F does not match the images. Please quantify wb data. Has statistical analysis been performed of the pERK p-peptides presented in the graph? What was the outcome?

The legend in Fig 4F (now 5F) has been modified to match the images. Western blots have been quantified and graphs of that quantification included. pERK is significantly higher in the M489V and the tg-AD (with or without M489V variant) compared to WT. We have included this in the text (p.11, lines 275-279).

'Immunoblot analysis revealed increased phosphorylation of ERK1/2 in brain lysates from both the M489V mice and the tg-AD mice compared to WT mice (**Figure 5F**). This increase was also captured by the phosphoproteomics analysis (**Figure 5G**). Thus, both the PKC α M489V mutation or the APP transgene enhance Erk signaling.'

7. Abeta analysis, please also provide IHC evidence of no change in abeta plaque pathology. These data and abeta ELISA data can be presented in Supp mat.

Fig. 6 with the abeta ELISA data have been moved to the Supplemental material (now **new Supplementary Figure 3A-C**); these results are now introduced earlier in the narrative (p.9, lines 221-229) (see comment from Reviewer 2). ICH analyses of WT tg-AD and M489V tg-AD brains were performed and included in the manuscript (**new Supplementary Figure 3D**). We have included this in the main text (p.9, lines 226-227):

"These results were confirmed by histochemistry analyses, where brains both WT tg-AD and M489V tg-AD mice presented similar levels of A β plaques (Supp Figure 3D)."

8. For e-phys measurements, to avoid the side effects of viral expression of CT100, please measure the e-phys properties of acute slices from the mice incubated with abeta.

The issue raised is whether viral expression, rather than CT100, is responsible for the observed effects. This has been addressed in a number of previous studies, including the original APP/transmission study, Kamenetz *et al.*⁹, and many subsequent others (including the Alfonso *et al.* Science Signaling study where we first showed that PKC α is necessary for A β -induced synaptic depression²). The consistent finding is that expression of GFP (or many other control proteins) with the same viral system has no effect on synaptic transmission^{2,9,10} or LTP¹¹.

9. Please describe in all the fig legends what statistical analysis have been used.

We have now included this in all the figure legends.

References:

1. Callender JA, *et al.* Protein kinase Calpha gain-of-function variant in Alzheimer's disease displays enhanced catalysis by a mechanism that evades down-regulation. *Proc Natl Acad Sci U S A* **115**, E5497-E5505 (2018).
2. Alfonso SI, *et al.* Gain-of-function mutations in protein kinase Calpha (PKCalpha) may promote synaptic defects in Alzheimer's disease. *Sci Signal* **9**, ra47 (2016).
3. Gustafsson B, Wigstrom H, Abraham WC, Huang YY. Long-term potentiation in the hippocampus using depolarizing current pulses as the conditioning stimulus to single volley synaptic potentials. *J Neurosci* **7**, 774-780 (1987).

4. Malinow R, Miller JP. Postsynaptic hyperpolarization during conditioning reversibly blocks induction of long-term potentiation. *Nature* **320**, 529-530 (1986).
5. Kelso SR, Ganong AH, Brown TH. Hebbian synapses in hippocampus. *Proc Natl Acad Sci U S A* **83**, 5326-5330 (1986).
6. O'Neill AK, *et al.* Protein kinase Calpha promotes cell migration through a PDZ-dependent interaction with its novel substrate discs large homolog 1 (DLG1). *J Biol Chem* **286**, 43559-43568 (2011).
7. Antal CE, *et al.* Cancer-associated protein kinase C mutations reveal kinase's role as tumor suppressor. *Cell* **160**, 489-502 (2015).
8. Tobias IS, Newton AC. Protein Scaffolds Control Localized Protein Kinase Czeta Activity. *J Biol Chem* **291**, 13809-13822 (2016).
9. Kamenetz F, *et al.* APP processing and synaptic function. *Neuron* **37**, 925-937 (2003).
10. Reinders NR, *et al.* Amyloid-beta effects on synapses and memory require AMPA receptor subunit GluA3. *Proc Natl Acad Sci U S A* **113**, E6526-E6534 (2016).
11. Hayashi Y, Shi SH, Esteban JA, Piccini A, Poncer JC, Malinow R. Driving AMPA receptors into synapses by LTP and CaMKII: requirement for GluR1 and PDZ domain interaction. *Science* **287**, 2262-2267 (2000).

REVIEWER COMMENTS

Reviewer #1 (Remarks to the Author):

The authors have adequately address my comments and I recommend that this manuscript is accepted.

Reviewer #2 (Remarks to the Author):

The authors have done an excellent job addressing my concerns, i have no further comments and support acceptance of the manuscript.

Reviewer #4 (Remarks to the Author):

The revised manuscript has addressed many of the 3 reviewers' comments, and improved the data quality. However, there are a few remaining concerns that need to be addressed regarding the significance and AD-relevance of this study. Given the lack of data with PKC inhibitors in the mouse experiments, and the scarce of support on PKC overactivity in human AD studies, the paper needs to tone down major statements.

1. The major basis of this paper is a GWAS identification of gain-of-function rare variants of PRKCA (encoding PKC α) in AD patients (ref. 21, Alfonso et al., 2016, Sci. Signal). Is it reproduced in other GWAS studies? It is also unclear whether other phosphoproteome studies (e.g. Bai et al., 2020, Neuron) have confirmed the increased phosphorylation of PKC substrates in AD patients. Ref. 30 can't be found in Pubmed.
2. The authors stated that "the relative abundance of PKC α did not change in brains from WT and M489V mice, ... M489V mutation in PKC α does not alter the steady-state levels of the protein (Fig. 1c)" (line 131-133). However, Fig. 1C shows a significant reduction of PKC α level in M489V mice. How to explain the inconsistency?
3. Fig. 7a shows a significant increase of PKC α level in AD patients, which is opposite to the finding in M489V mice (Fig. 1c). Moreover, is the increase of PKC α in AD supported by proteome studies?
4. The title needs to be toned down. It is very unlikely that cognitive decline in AD is driven by a modest increase of PKC activity. At most the paper only shows cognitive decline in a mutant mouse line expressing PKC α M489V.
5. The electrophysiological data (Fig. 4) are rather limited. Prior reviewer's suggestion of checking the impact of PKC α M489V on synaptic transmission and synaptic plasticity was not addressed experimentally. Does PKC α M489V affect baseline synaptic currents (control-WT vs. control-M489V)? Fig. 4c used normalized EPSCs (controls in both groups were normalized to 1), but compared CT100-infected WT vs. M489V, which is incorrect and needs to be re-analyzed (using paired t-test is not right here). Fig. 4d, the electrophysiological traces of controls in WT vs. M489V have very different amplitudes (shown in scale bars), and CT100-infected-M489V actually has a larger size than CT100-infected-WT, which is opposite to the statistics shown in Fig. 4c (individual data points need to be added). Fig. 4a should use a real image instead of a drawing.
6. Fig. 2a shows the modest reduction of spine density in M489V mice (4-5 months old), and the authors stated that it was "neurite degeneration" (line 171). However, without knowing whether the initial spine density is normal, and whether the change is age-related, the conclusion is an inaccurate overstatement.
7. Behavioral data in Fig. 3 and Fig. 6 have largely redundant information between WT v. M489V. They should be consolidated to avoid redundancy.

8. Many figures do not have individual dots on the bar graph. Authors stated in the response that the data points obscure the bars and error bars. Using gray (non-filled) dots should be able to solve this problem. Raw data values and detailed statistics requested by the journal (source data excel files) are not included in the submission.

Reviewer #4 (Remarks to the Author):

The revised manuscript has addressed many of the 3 reviewers' comments and improved the data quality. However, there are a few remaining concerns that need to be addressed regarding the significance and AD-relevance of this study. Given the lack of data with PKC inhibitors in the mouse experiments, and the scarce of support on PKC overactivity in human AD studies, the paper needs to tone down major statements.

1. The major basis of this paper is a GWAS identification of gain-of-function rare variants of PRKCA (encoding PKC α) in AD patients (ref. 21, Alfonso et al., 2016, Sci. Signal). Is it reproduced in other GWAS studies?

This is not a GWAS identification, and we apologize for the confusion. We have rewritten the text to emphasize that **the M489V variant is an exceedingly rare mutation which we have identified using whole genome sequencing data in a large AD family-based cohort from NIMH**. Recent large GWAS have not reported this mutation; however, GWAS focus mostly on common variants which can be genotyped or reliably imputed.

We have edited the text in the Introduction for clarity on this issue (lines 59-63):

“In a recent search for rare functional variants associated with AD, analysis of whole genome sequencing data from 410 families of affected and unaffected siblings from the NIMH cohort identified variants in PKC α . One of these variants, M489V (rs34406842, minor allele frequency of 0.00095 in gnomad), was present only in affected members and no unaffected members of 4 families, cosegregating with AD affection status.”

It is also unclear whether other phosphoproteome studies (e.g. Bai et al., 2020, Neuron) have confirmed the increased phosphorylation of PKC substrates in AD patients. Ref. 30 can't be found in Pubmed.

Here is the link to Reference 30:

<https://www.nature.com/articles/s43587-021-00071-1>

In the phosphoproteomic studies we cited, **elevated PKC signaling was reported as the #1 upregulated signaling pathway**. Elevated PKC signaling is also apparent in the Bai et al. study the reviewer asks about, although the authors of this paper focus on Abeta-associated pathways. Nonetheless, they identify PKC as one of the main kinases 'likely activated in AD'. Furthermore, when we checked their dataset, we saw that the phosphorylation levels of the turn motif of PKC α , an indicator of PKC levels, is increased in the Alzheimer's disease brains. We have added this reference to the manuscript and rewritten this sentence in the Discussion (lines 350-355):

“A comprehensive phosphoproteome analysis by Tagawa et al. revealed that PKC substrates account for over half of the core molecules that display with increased phosphorylation in AD brains, a finding supported by subsequent phosphoproteomics studies identifying PKC as one of the main kinases activated in AD, and reporting increased phosphorylation of PKC α at Thr638, a quantitatively phosphorylated C-terminal site that serves as indicator of PKC steady state levels.”

2. The authors stated that “the relative abundance of PKC α did not change in brains from WT and M489V mice, ... M489V mutation in PKC α does not alter the steady-state levels of the protein (Fig. 1c)” (line 131-133). However, Fig. 1C shows a significant reduction of PKC α level in M489V mice. How to explain the inconsistency?

We thank the reviewer for pointing out that, in fact, the small change in PKC α levels (8%) in Figure 1C gained statistical significance. In every other analysis, there was no difference (including the WT mice on the B6;SJL background(non-tg and tg AD mice), see Figure 5B). Additionally, there was no difference in the PKC α protein levels *in the mice used for the phosphoproteomics* measured by more rigorous Western Blot analysis; we had included the Western Blot analysis in our earlier PNAS paper ¹ (Callender *et al.* 2018; ref 34 in the paper). We have now moved Figure 1C data to (new) Supplementary Figure 1H and added also the relative abundance data for the 6-month cohort, which shows equal levels in both WT and M489V mice (source data of proteomics analysis from this cohort can be found in Baffi *et al.* 2021²). We have rewritten the sentence as follows (lines 135-141):

*“Importantly, immunoblot analysis of brain lysates from littermates of the mice used for mass spectrometry analysis has previously established that the amount of PKC α protein did not change in brains from WT and M489V mice (see also Supp **Figure 1H**), consistent with our biochemical studies which demonstrated that the M489V mutation in PKC α does not alter the steady-state levels of the protein”.*

3. Fig. 7a shows a significant increase of PKC α level in AD patients, which is opposite to the finding in M489V mice (Fig. 1c). Moreover, is the increase of PKC α in AD supported by proteome studies?

The mouse data are completely consistent with the human data: *more PKC α signaling in both cases*. In our mouse study, PKC α signaling was enhanced because of a mutation that *increased the catalytic rate of the enzyme*. In humans that have the M489V variant, PKC α signaling will be enhanced because the enzyme is more active. But enhanced PKC α signaling can occur by an abundance of different mechanisms that control PKC α , including the many mechanisms that control its steady-state levels (kinases that control its priming phosphorylations, mTORC2 and PDK1; phosphatases that control its degradation such as PHLPP; Pin1, E3 ligases, etc). Because PKC is haploinsufficient, the amount in the cell is important. Higher steady-state levels result in higher signaling output.

We have revised the text to better explain that the amount of PKC regulates its signaling output (lines 320-323):

“In this regard, the amount of PKC protein in cells regulates its signaling output ³, with higher steady-state levels resulting in higher signaling output; for example, the related isozyme, PKC β II, has been shown to be haplo-insufficient in suppressing oncogenic signaling ⁴.”

And in Discussion (lines 369-373):

“The steady-state levels of PKC are precisely set by diverse mechanisms, including stabilizing phosphorylations mediated by mTORC1 and PDK-1 and opposed by the quality control phosphatase PHLPP, as well as regulators such as Pin1, E3 ligases, and heat shock proteins ⁵. This suggests that not only are mutations in PKC α biomarkers for the disease, but the intrinsic set point of PKC α levels, controlled by diverse regulators, may also predict susceptibility to AD.”

4. The title needs to be toned down. It is very unlikely that cognitive decline in AD is driven by a modest increase of PKC activity. At most the paper only shows cognitive decline in a mutant mouse line expressing PKC α M489V.

We agree that most readers will also be surprised by this finding, which is the most striking take home message from our work: a very small change in a protein kinase, substitution of a Met for a Val, results in a pathophysiology. There are many other examples of small changes in the activity of kinases causing disease. Most notable is spinocerebellar ataxia type 14, where very modest changes in the autoinhibition of another PKC family member, the brain-specific PKC gamma, are causative in the disease^{6,7,8}. These disease mutations underscore how the activity of PKC family members is exquisitely tuned and this is essential for homeostasis. Any aberrant activity leads to pathophysiologies, even seemingly 'small' changes.

The same is the case for other kinases. Perhaps most relevant is Protein Kinase A, in the same AGC family as PKC: single amino acid germline variants that cause similarly subtle (<30%) increases in catalytic activity cause congenital malformation syndrome (PMCID: PMC7675002) and Cushing's disease (PMCID: PMC6713507).

Regarding our own study, the only difference in our mice is this one amino acid change. We show this was sufficient to rewire the brain phosphoproteome and cause cognitive decline. Our previous study established the genetics with affected individuals in the same family having the mutation and unaffected not having it. Hopefully the rewritten explanation of the genetics in the Introduction (see 1. above) will make this clear. Additionally, we have revised the title as follows:

"Enhanced Activity of Alzheimer Disease-associated Variant of Protein Kinase C α Drives Cognitive Decline in a Mouse Model"

5. The electrophysiological data (Fig. 4) are rather limited. Prior reviewer's suggestion of checking the impact of PKC α M489V on synaptic transmission and synaptic plasticity was not addressed experimentally. Does PKC α M489V affect baseline synaptic currents (control-WT vs. control-M489V)?

This is an interesting question, but the question addressed in Figure 4 is **whether A β has a greater effect on synaptic transmission in PKC α -M489V animals**. For this question, one needs to measure the A β -induced synaptic depression produced in tissue from WT animals and compare it to the A β -induced synaptic depression produced in tissue from PKC α -M489V animals. The paired recording method is the only way to address this question. The conclusion of the data presented in Figure 4 is that synaptic transmission in animals harboring the PKC α M489V mutation is more sensitive to A β (i.e. have more reduced transmission in the presence of elevated A β). It is one of the most well-established concepts in the synaptic plasticity field that reduced transmission will reduce activation of NMDA-receptors during periods of plasticity, and thus reduce plasticity such as LTP, which is responsible for learning and memory. We have consulted at length with our colleague Dr. Roberto Malinow who is an expert in this area (see Acknowledgements).

Fig. 4c used normalized EPSCs (controls in both groups were normalized to 1), but compared CT100-infected WT vs. M489V, which is incorrect and needs to be re-analyzed (using paired t-test is not right here).

In order to compare synaptic depression produced by elevated A β (i.e. by expression of CT100), transmission in each genotype needs to be normalized to the non-infected cell. This is the principle behind dual-patch recordings, an approach that is widely used and permits direct comparison of synaptic transmission onto a cell expressing 'protein X' to synaptic transmission onto cells not expressing 'protein X'. We added text in the results section to better describe the validity of this approach, see lines 212-215.

“Such cell-pair recordings permit one to compare directly the impact of elevated A β on synaptic transmission, as the number of activated Shaffer collateral axons targeting infected and non-infected cells is on average the same, irrespective of the stimulation intensity.”

Fig. 4d, the electrophysiological traces of controls in WT vs. M489V have very different amplitudes (shown in scale bars), and CT100-infected-M489V actually has a larger size than CT100-infected-WT, which is opposite to the statistics shown in Fig. 4c (individual data points need to be added).

The different amplitudes of transmission onto non-infected neurons (in WT and PKC α -M489V tissue) is simply because different numbers of Shaffer collateral axons were stimulated. But on average the same number of axons will be activated targeting infected and non-infected neurons. This permits comparison of transmission onto infected and non-infected neurons. On average, for this experiment, we observed very similar responses amplitudes, see graph below.

Individual raw data points (not normalized) are shown in the dot plot presented in Figure 4B. It may be redundant to show them again in Figure 4C; nonetheless, we have now added them in Figure 4C.

Fig. 4a should use a real image instead of a drawing.

We agree that Fig 4A is not adequate. We have modified it with a diagram (**new Fig 4A**) that demonstrates why paired recordings are necessary: a stimulating electrode activates an

unknown number of Shaffer collateral axons. But on average, the same number will target the infected and non-infected neuron. A real image of a brain slice would not display the importance of paired recordings.

6. Fig. 2a shows the modest reduction of spine density in M489V mice (4-5 months old), and the authors stated that it was “neurite degeneration” (line 171). However, without knowing whether the initial spine density is normal, and whether the change is age-related, the conclusion is an inaccurate overstatement.

We have reworded the sentence to state (line 178):

“In summary, the PKC α M489V variant mice display reduced spine density, as well as enhanced phosphorylation of proteins that regulate neurites, an important hallmark of Alzheimer’s disease.”

7. Behavioral data in Fig. 3 and Fig. 6 have largely redundant information between WT v. M489V. They should be consolidated to avoid redundancy.

Figure 3 presents the data of the effect of the M489V mutation on WT mouse (C57BL/6) whereas Figure 6 presents it on the background of the APPSWE mouse (B6;SJL). For rigor, we also included side by side the analysis of the control mouse for the APPSWE line (i.e. non-tg AD, B6;SJL mouse). We prefer to keep this level of rigor as it also demonstrates that the effects are not strain dependent.

8. Many figures do not have individual dots on the bar graph. Authors stated in the response that the data points obscure the bars and error bars. Using gray (non-filled) dots should be able to solve this problem. Raw data values and detailed statistics requested by the journal (source data excel files) are not included in the submission.

Point well taken. We have replaced graphs with ones showing individual data points throughout the manuscript, as suggested by the reviewer. Additionally, we have also uploaded an excel file with all the raw data values and a file with all the original gels from the western blots.

REFERENCES

1. Callender JA, *et al.* Protein kinase Calpha gain-of-function variant in Alzheimer's disease displays enhanced catalysis by a mechanism that evades down-regulation. *Proc Natl Acad Sci U S A* **115**, E5497-E5505 (2018).
2. Baffi TR, *et al.* mTORC2 controls the activity of PKC and Akt by phosphorylating a conserved TOR interaction motif. *Sci Signal* **14**, (2021).
3. Newton AC. Protein kinase C: perfectly balanced. *Crit Rev Biochem Mol Biol* **53**, 208-230 (2018).

4. Antal CE, *et al.* Cancer-associated protein kinase C mutations reveal kinase's role as tumor suppressor. *Cell* **160**, 489-502 (2015).
5. Baffi TR, Newton AC. Protein kinase C: release from quarantine by mTORC2. *Trends Biochem Sci* **47**, 518-530 (2022).
6. Wong MMK, *et al.* Neurodegeneration in SCA14 is associated with increased PKCgamma kinase activity, mislocalization and aggregation. *Acta Neuropathol Commun* **6**, 99 (2018).
7. Verbeek DS, Knight MA, Harmison GG, Fischbeck KH, Howell BW. Protein kinase C gamma mutations in spinocerebellar ataxia 14 increase kinase activity and alter membrane targeting. *Brain* **128**, 436-442 (2005).
8. Pilo CA, *et al.* Protein Kinase C γ Mutations Drive Spinocerebellar Ataxia Type 14 by Impairing Autoinhibition. *bioRxiv*, 2021.2006.2024.449810 (2021).

REVIEWER COMMENTS

Reviewer #4 (Remarks to the Author):

The re-revised paper has addressed most of my prior concerns, and improved the presentation. A few minor concerns need to be addressed.

1. Fig. 3 and its legend are inconsistent.

2. Fig. 4c should use two-way ANOVA with post hoc comparisons instead of t-test, since the comparisons are not limiting to two groups. Only WT-control should be normalized to 1, so WT+CT100 vs. M489V+CT100 can be compared.

3. Fig. 4d should select representative examples that do not show significant differences between WT-control vs. M489-control (as shown in the rebuttal figure). The current traces are hugely different (scale bar: 6pA vs. 20pA). The black vs. gray traces need to be labeled clearly.

Reviewer #4 (Remarks to the Author):

The re-revised paper has addressed most of my prior concerns, and improved the presentation. A few minor concerns need to be addressed.

1. Fig. 3 and its legend are inconsistent.

Thanks for bringing this to our attention. Figure 3 and its legend are now properly labeled/described.

2. Fig. 4c should use two-way ANOVA with post hoc comparisons instead of t-test, since the comparisons are not limiting to two groups. Only WT-control should be normalized to 1, so WT+CT100 vs. M489V+CT100 can be compared.

To assess the statistical significance of dual-patch recordings made in cell pairs, a paired T-test is the best choice and was used to compare CT100-infected cells with control cells. However, we agree that to compare WT+CT100 with M489V+CT100, a two-way ANOVA test followed with a multiple comparisons post hoc test is more appropriate. More information about the exact tests used is included in the Methods and in the legend for Figure 4 (lines 623-625, and 1052-1053). Importantly, both WT-control and PKC α -M489V-control groups need to be normalized to 1 as this is the principle behind dual patch recordings, as we explained in the previous response letter of Aug 2nd 2022:

In order to compare synaptic depression produced by elevated A β (i.e. by expression of CT100), transmission in each genotype needs to be normalized to the non-infected cell. This is the principle behind dual-patch recordings, an approach that is widely used and permits direct comparison of synaptic transmission onto a cell expressing 'protein X' to synaptic transmission onto cells not expressing 'protein X'. We added text in the results section to better describe the validity of this approach, see lines 212-215.

"Such cell-pair recordings permit one to compare directly the impact of elevated A β on synaptic transmission, as the number of activated Shaffer collateral axons targeting infected and non-infected cells is on average the same, irrespective of the stimulation intensity."

3. Fig. 4d should select representative examples that do not show significant differences between WT-control vs. M489-control (as shown in the rebuttal figure). The current traces are hugely different (scale bar: 6pA vs. 20pA). The black vs. gray traces need to be labeled clearly.

We changed the example traces for the WT mice recordings and the scale is now the same between the two genotypes. The traces are now labeled, thank you for bringing this to our attention.

REVIEWERS' COMMENTS

Reviewer #4 (Remarks to the Author):

The authors have addressed all concerns. The paper is now suitable for publication.

Reviewer #4 (Remarks to the Author):

The authors have addressed all concerns. The paper is now suitable for publication.

We have no comments since all concerns were addressed.